# Synaptic memory requires CaMKII

**Wucheng Tao[1,2], Joel Lee[2†], Xiumin Chen[2†], Javier Díaz-Alonso[2‡], Jing Zhou[3], Samuel Pleasure[3], Roger A Nicoll[2,4]***

[1]Key Laboratory of Brain Aging and Neurodegenerative Diseases, Fujian Medical University, Fuzhou, China; [2]Department of Cellular and Molecular Pharmacology, University of California, San Francisco, San Francisco, United States; [3]Department of Neurology, University of California, San Francisco, San Francisco, United States; [4]Physiology, University of California, San Francisco, San Francisco, United States

**Abstract** Long-term potentiation (LTP) is arguably the most compelling cellular model for learning and memory. While the mechanisms underlying the induction of LTP ('learning') are well understood, the maintenance of LTP ('memory') has remained contentious over the last 20 years. Here, we find that $Ca^{2+}$-calmodulin-dependent kinase II (CaMKII) contributes to synaptic transmission and is required LTP maintenance. Acute inhibition of CaMKII erases LTP and transient inhibition of CaMKII enhances subsequent LTP. These findings strongly support the role of CaMKII as a molecular storage device.

## Editor's evaluation

The article addresses the 'decades-old' unresolved question as to whether CaMKII is required for the maintenance of synaptic long-term potentiation, and shows, based on a set of elegant experiments, that it indeed is.

*For correspondence:
roger.nicoll@ucsf.edu

†These authors contributed equally to this work

Present address: ‡Department of Neuroscience, University of California, Irvine, United States

**Competing interest:** The authors declare that no competing interests exist.

## Introduction

The mechanism by which the brain stores information has fascinated neuroscientists for well over a century. Two general ideas have emerged. The first proposes that information is held by ongoing activity in neuron ensembles. This, indeed, underlies short-term working memory (***D'Esposito and Postle, 2015***; ***Inagaki et al., 2019***; ***Wang, 2001***). However, it is clear that such a mechanism cannot account for long lasting, enduring memories. For instance, it is well established that patients who recover from prolonged periods of isoelectric EEG and brain stem silence following barbiturate overdose have no detectable cognitive deficits, for example, ***Bird and Plum, 1968***. The second and generally accepted view, first enunciated by Cajal over a century ago (***Cajal, 1911***), is the modification of neuronal connections. How these changes might be stored has remained a mystery.

Over the past few decades, $Ca^{2+}$-calmodulin-dependent kinase II (CaMKII) has emerged as a very attractive molecular candidate for information storage (***Bayer and Schulman, 2019***; ***Bhattacharyya et al., 2020***; ***Hell, 2014***; ***Kennedy, 2013***; ***Lisman et al., 2002***; ***Lisman et al., 2012***; ***Lisman and Goldring, 1988***). This dodecameric kinase is activated by $Ca^{2+}$/CaM resulting in autophosphorylation and the activity of the kinase remains after the removal of $Ca^{2+}$ (***Miller and Kennedy, 1986***), a state referred to as autonomy. Furthermore, evidence suggests that the enzyme can undergo activation-triggered subunit exchange such that subunits that did not experience the original $Ca^{2+}$ transient are phosphorylated, a mechanism postulated to allow for continued phosphorylation in the face of protein turnover (***Bhattacharyya et al., 2020***; ***Bhattacharyya et al., 2016***; ***Stratton et al., 2014***) Thus, CaMKII has many attractive biochemical features expected for a memory molecule. What is the physiological evidence for such a role?

**eLife digest** How the brain stores information is a question that has fascinated neuroscientists for well over a century. Two general ideas have emerged. The first is that groups of neurons hold information by staying active. The second is that they hold information by strengthening their connections to one another, making it easier for them to work together in the future. Scientists call this second idea 'long-term potentiation'.

One of the molecules involved in long-term potentiation is a protein called calcium-calmodulin-dependent kinase II, or CaMKII for short. Blocking CaMKII, or deleting its gene, stops the connections between neurons from becoming stronger. This suggests neurons need CaMKII to learn, but it remains unclear whether neurons also use CaMKII to maintain neuronal memories after they have been created. If CaMKII does play a role in maintaining memories, blocking it after learning should reverse the learning process, but so far, experiments have not been able to show this.

Tao et al. revisited these experiments to find out more. They examined slices of brain tissue from mice that had been treated with fast-acting CaMKII inhibitors. It took tens of minutes, but the inhibitors were able to reverse long-term potentiation, both for newly acquired neuronal memories and for older memories that had formed when the mice were alive. The choice of CaMKII inhibitor and the time lag could explain why scientists have not observed the effect before.

Understanding long-term potentiation is a fundamental part of understanding learning and memory. It could also reveal more about the opposite phenomenon: long-term depression. This is a type of learning where the connections between neurons become weaker. Long-term depression also takes tens of minutes to occur, suggesting that future research into CaMKII might shed light on how it works.

Long-term potentiation (LTP), in which brief high-frequency synaptic stimulation results in a lasting increase in synaptic strength, is an attractive cellular model for learning and memory (*Choquet, 2018*; *Collingridge et al., 2004*; *Huganir and Nicoll, 2013*; *Malinow and Malenka, 2002*; *Nicoll, 2017*). A role for CaMKII in LTP is compelling. Pharmacologically blocking CaMKII (*Malenka et al., 1989*; *Malinow et al., 1989*; *Otmakhov et al., 1997*) or genetically deleting CaMKII (*Giese et al., 1998*; *Incontro et al., 2018*; *Silva et al., 1992*) blocks LTP and constitutively active CaMKII both mimics and occludes LTP (*Lledo et al., 1995*; *Pettit et al., 1994*; *Pi et al., 2010*; *Poncer et al., 2002*). However, despite the attractive biochemical properties of CaMKII, whether CaMKII is involved in the *maintenance* of LTP and, by extension, memory storage remains problematic.

There are two predictions if CaMKII is responsible for LTP maintenance and synaptic memory (*Lisman, 2017*; *Sanhueza and Lisman, 2013*). First, transiently blocking CaMKII *after* the induction of LTP should cause a long-lasting erasure. Attempts at reversing LTP have failed on numerous occasions (*Buard et al., 2010*; *Chen et al., 2001*; *Malinow et al., 1989*; *Murakoshi et al., 2017*; *Otmakhov et al., 1997*), but see *Feng, 1995*. Second, if CaMKII is involved in synaptic memory, one would expect it to leave a lasting trace at synapses (*Lisman, 2017*; *Sanhueza et al., 2011*; *Sanhueza and Lisman, 2013*). Thus, silencing CaMKII should reduce synaptic transmission. Interestingly, the CaMKII/NMDAR complex does exist under basal conditions (*Leonard et al., 1999*). However, the genetic deletion of CaMKII (*Achterberg et al., 2014*; *Giese et al., 1998*; *Silva et al., 1992*), but see *Hinds et al., 1998*, or pharmacological inhibition of CaMKII does not affect baseline synaptic responses (*Buard et al., 2010*; *Chen et al., 2001*; *Feng, 1995*; *Malinow et al., 1989*; *Murakoshi et al., 2017*; *Otmakhov et al., 1997*; *Wang and Kelly, 1996*). Finally, using a FRET-based CaMKII sensor, CaMKII activation in spines after LTP induction persists for only ~1 min (*Lee et al., 2009*). Thus, physiological support for a role of CaMKII in maintaining LTP or synaptic transmission is lacking.

Recent findings have prompted us to reevaluate CaMKII's role in the maintenance of LTP. Deleting CaMKII with CRISPR (*Incontro et al., 2018*) or expressing a peptide inhibitor of CaMKII (*Goold and Nicoll, 2010*; *Sanhueza et al., 2011*) cause a substantial reduction in synaptic transmission. Furthermore, studies using a membrane permeable peptide inhibitor of CaMKII (tatCN21 or antCN27) have reported a lasting depression in synaptic transmission (*Barcomb et al., 2016*; *Gouet et al., 2012*; *Sanhueza et al., 2011*; *Sanhueza et al., 2007*) and evidence for a reduction in LTP maintenance (*Sanhueza et al., 2011*; *Sanhueza et al., 2007*).

To reevaluate the role of CaMKII in synaptic memory we have used two different rapidly acting and reversible CaMKII inhibitors. As would be expected if LTP contributes to synaptic memory, inhibition of CaMKII depresses synaptic transmission and the properties of this persistent action of CaMKII are remarkably similar to those of LTP. Furthermore, we demonstrate that applying these inhibitors *after* inducing LTP fully reverses LTP, indicating that CaMKII is required for the persistence of LTP. Thus, our findings strongly support CaMKII as a molecular storage device. Reasons for why the present results differ from most previous studies are discussed.

## Results

### Blocking CaMKII inhibits synaptic transmission

Previous studies established, both with the expression of inhibitory peptides (*Goold and Nicoll, 2010*; *Sanhueza et al., 2011*) or CRISPR deletion of CaMKII (*Incontro et al., 2018*) that CaMKII contributes ~50% to AMPAR responses. In order to study the basis for this activity as well as the role of CaMKII in LTP maintenance, rapid and reversible block of CaMKII is essential. Since peptide inhibitors have proved most effective, this requires making these peptides membrane permeable either by cell-penetrating peptides (CPPs) (e.g., tat) or by protein lipidation (e.g., myristoylation). Delivery of peptides into cells has profound therapeutic potential and therefore has received a great deal of study (*Allen et al., 2018*; *LeCher et al., 2017*; *Nelson et al., 2007*; *Patel et al., 2019*). The effectiveness of CPPs is hotly debated, because, although these peptides clearly enter the cell by endocytosis, it is not certain to what degree these peptides actually have access to the cytoplasm. This appears not to be an issue with lipid modification.

As an initial assay we bathed acute slices in a peptide inhibitor and measured the AMPAR/NMDAR ratio. We would expect the ratio to be reduced by ~50%, since expressing these peptides had no effect on NMDAR currents. We first tested the effectiveness of tatCN21 (5 µM) (*Barcomb et al., 2016*; *Buard et al., 2010*; *Sanhueza et al., 2011*) in reducing the AMPAR/NMDAR ratio (*Figure 1—figure supplement 1*). In agreement with previous reports, we failed to see an effect on AMPAR responses. Since higher concentrations of tatCN21 have significant nonspecific effects (*Barcomb et al., 2016*), we turned to other peptides. Previous work (*Goold and Nicoll, 2010*) showed that prolonged (1–2 days) incubation with myr-CN27 (10 µM) caused a ~50% reduction in the AMPAR/NMDAR ratio. We, therefore, repeated this experiment with shorter incubations (1–2 hr) and a lower concentration (1 µM) and found that myr-CN27 reduced the ratio by ~50% (*Figure 1—figure supplement 1*). What might account for the lack of effect of tatCN21? To address this, we repeated the experiment using myr-CN21 (5 µM) and found that it reduced the ratio by ~50% (*Figure 1—figure supplement 1*), suggesting that lipidation is a more effective mode of delivery than CPPs. We, therefore, selected myr-CN27 for our studies.

Application of myr-CN27 to acute slices (1 µM) caused a highly reproducible slowly developing depression of synaptic AMPAR mediated EPSCs, which stabilized after approximately 40 min (*Figure 1A, B*). A number of experiments were carried out to ensure that the effect we observe with myr-CN27 is due to the specific inhibition of CaMKII. First, in many experiments we measured the size of the NMDAR EPSC, by depolarizing the cell to +40 mV before and after the application of the peptide. We found that myr-CN27 had no effect on the NMDAR EPSC (*Figure 1A*, red circles, *Figure 1B*). Second, it is important to note that the magnitude of the decrease in transmission seen with myr-CN27 is similar to that reported with the genetic deletion of CaMKIIα (*Incontro et al., 2018*) suggesting that myr-CN27 fully silences CaMKII. Finally, we examined the effects of myr-CN27 in cells lacking CaMKIIα (*Figure 1C, D*). CaMKIIα was deleted using CRISPR (*Incontro et al., 2018*). We simultaneously recorded from control cells and cells lacking CaMKIIα in slice culture. In the absence of CaMKIIα (green circle) synaptic responses were ~50% of control (black circles), as expected. Application of myr-CN27 depressed controls cells to the level recorded in cells lacking CaMKIIα (*Figure 1C, D*, black circles), but it had no additional effect on cells lacking CaMKIIα (green circles) (*n* = 6). These results demonstrate that the effect of CN27 is fully explained by its selective and complete silencing of CaMKIIα.

Previous studies have relied on field potential recording and thus we repeated our experiments with this approach (*Figure 1E*). We still recorded a depression, but in this case the depression took tens of minutes to develop (black symbols). We interleaved these experiments with whole cell recording

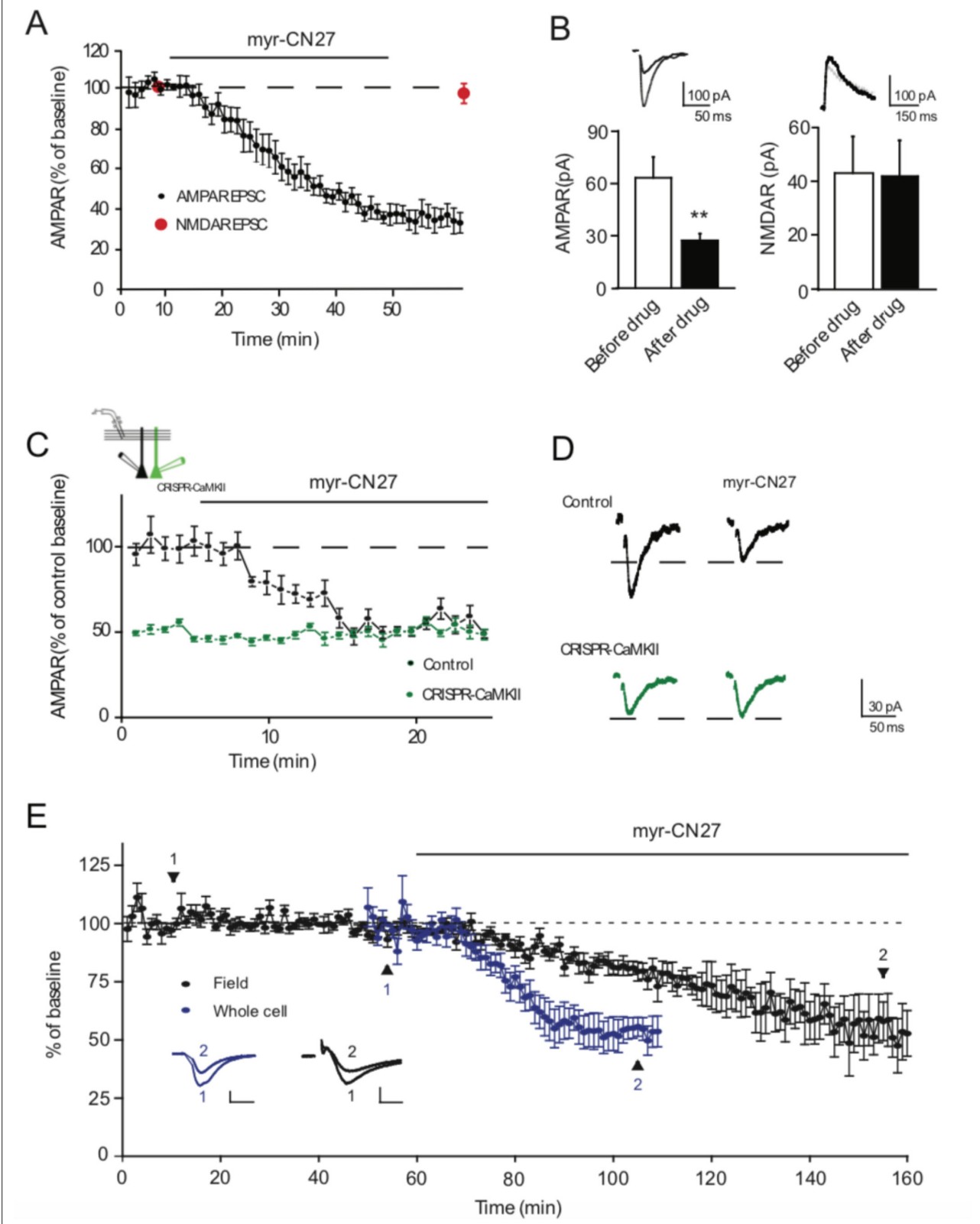

**Figure 1.** Ca²⁺-calmodulin-dependent kinase II (CaMKII) inhibitor myr-CN27 inhibits AMPAR synaptic transmission through CaMKIIα. (**A**) Time course of myr-CN27 (1 μM) inhibition on AMPAR EPSCs and NMDAR EPSCs in acute slices. (**B**) Summary graphs showing myr-CN27 inhibited AMPAR EPSCs (before drug: 63.9 ± 12 pA; after drug: 27.6 ± 3.9 pA; $n$ = 10, $p < 0.01$, two-tailed Wilcoxon Signed Rank Test), but not NMDAR EPSCs (before drug: 43.2 ± 13.7 pA; after drug: 42 ± 13.4 pA; $n$ = 5, $p > 0.05$, two-tailed Wilcoxon Signed Rank Test). (**C**) Time course of the effect of myr-CN27 on AMPAR EPSCs

*Figure 1 continued on next page*

*Figure 1 continued*

in wt cells and simultaneously recorded CRISPR-CaMKIIα transfected cells, normalized to wt baseline (from culture slices). While myr-CN27 inhibited AMPAR EPSCs in wt cells, it had no effect on CRISPR-CaMKIIα transfected cells ($n = 6$, $p > 0.05$, two-tailed Wilcoxon Signed Rank Test). (**D**) Sample traces showing that myr-CN27 inhibited AMPAR EPSCs in control cells, but had no effect in CRISPR-CaMKIIα transfected cells. Black traces are control cell, green traces are transfected cell. Mean ± standard error of the mean (SEM). \*\*$p < 0.01$. (**E**) A comparison of the reduction of synaptic transmission in field and whole cell recording. For the field recording (black, $n = 8$) after a 60-min stable baseline, 1 µM myr-CN27 was applied to the slice for 100 min. For the whole cell recording (blue, $n = 7$), the drug was applied after a 10-min stable baseline. In both, the response was reduced to 50% of baseline. Example traces for both are shown taken at the timepoints indicated by 1 and 2. The field EPSPs shown on the bottom right have scale bars vertical 0.1 mV and 5 ms horizontal. The whole cell recording shown bottom left have scale bars 50 pA vertical and 20 ms horizontal. Both recordings are done in WT slices with the whole cell recordings done in slice culture DIV3–14 while the fields were done in acute slices from P15–28 mice.

The online version of this article includes the following source data and figure supplement(s) for figure 1:

**Source data 1.** Ca$^{2+}$-calmodulin-dependent kinase II (CaMKII) inhibitor myr-CN27 inhibits AMPAR synaptic transmission through CaMKIIα.

**Figure supplement 1.** The effect of various Ca$^{2+}$-calmodulin-dependent kinase II (CaMKII) peptide inhibitors on the AMPAR/NMDAR ratio.

and confirmed that the depression is considerably faster (blue symbols). We interpret the difference in time course to the fact that whole cell recordings are made from superficial cells, while field potential responses are generated by populations of neurons throughout the depth of the slice. Penetration of peptides into the slice is expected to be slow.

We next examined the effects of AIP, a peptide inhibitor designed from the autoinhibitory domain. We first expressed AIP in neurons in slice culture and compared AMPAR and NMDAR responses to those in simultaneously recorded neighboring control cells (***Figure 2A***). Similar to the effect of expressing CN27 (***Goold and Nicoll, 2010***) or applying myr-CN27 (see above), there was a selective ~50% reduction in the AMPAR EPSC, but no change in the NMDAR response. Given these positive results, we bathed acute slices in myr-AIP (20 µM) and measured the AMPAR/NMDAR ratio (***Figure 2B***) ($n = 12$). The ratio was significantly reduced. Acute application of myr-AIP in slice culture caused a slowly developing depression in synaptic activity (***Figure 2C***, black circles) ($n = 6$) and this effect is due to the selective inhibition of CaMKII, because interleaved recording from cells in which CaMKIIα had been deleted (***Figure 2C***, green circles), myr-AIP no longer had any effect ($n = 5$). These findings show that myr-CN27 and myr-AIP selectively and completely inhibit CaMKII, both in acute slices and in slice culture. It is interesting that, while CaMKII is expressed presynaptically and reported to effect transmitter release (***Benfenati et al., 1992***; ***Hinds et al., 2003***), there was no presynaptic effect of the peptide inhibitors.

The time course of inhibition for both myr-CN27 and myr-AIP is slow, requiring 10's of minutes. Is this due to the slow access of these peptides to the cell interior, or is it due to the slow reversal of the action of CaMKII? To address this, we turned to a novel light inducible inhibitor of CaMKII, in which the inhibitory peptide AIP2 (***Ishida et al., 1998***) is linked to the LOV-Jα helix domain of Phototropin (referred to as paAIP2) (***Murakoshi et al., 2017***). Blue light rapidly exposes the AIP2 peptide, which then returns to its closed inactive conformation in approximately 40 s following light exposure. We expressed paAIP2 in individual neurons with biolistics, which, unlike the myristoylated peptides, has the advantage of limiting the manipulation to individual postsynaptic cells. The synaptic properties of neurons expressing paAIP2 were not different from the properties of neighboring control cells (***Figure 3—figure supplement 1A, B***) ($n = 16$). Exposure of these cells to blue light resulted in a slowly developing inhibition of AMPAR responses (***Figure 3A***) ($n = 12$), similar to the kinetics of myr-CN27 and myr-AIP. The blue light had no effect on simultaneously recorded control cells (***Figure 3A***, black circles) or in cells coexpressing paAIP2 with CRISPR to delete CaMKIIα (***Figure 3—figure supplement 2***) ($n = 5$), establishing the specificity of paAIP2. These results with paAIP2 suggest that the slow time course is not due to slow access, but rather due to the slow reversal of the action of CaMKII.

## The origin of the constitutive action of CaMKII

The persistent CaMKII activity has two possible origins. First, it could represent ongoing constitutive Ca$^{2+}$ activation of CaMKII from various sources. It seems unlikely that resting Ca$^{2+}$ levels (***Maravall et al., 2000***) stimulate CaMKII, since the sensitivity of CaMKII to Ca$^{2+}$ requires considerably higher levels of Ca$^{2+}$ (***Schulman, 1984***). To directly test whether constitutive CaMKII activity is Ca$^{2+}$ dependent, we loaded cells with the Ca$^{2+}$ chelator BAPTA (15 mM). However, myr-CN27 retained its depressive effect (***Figure 4—figure supplement 1A***). This finding also rules out spontaneous background

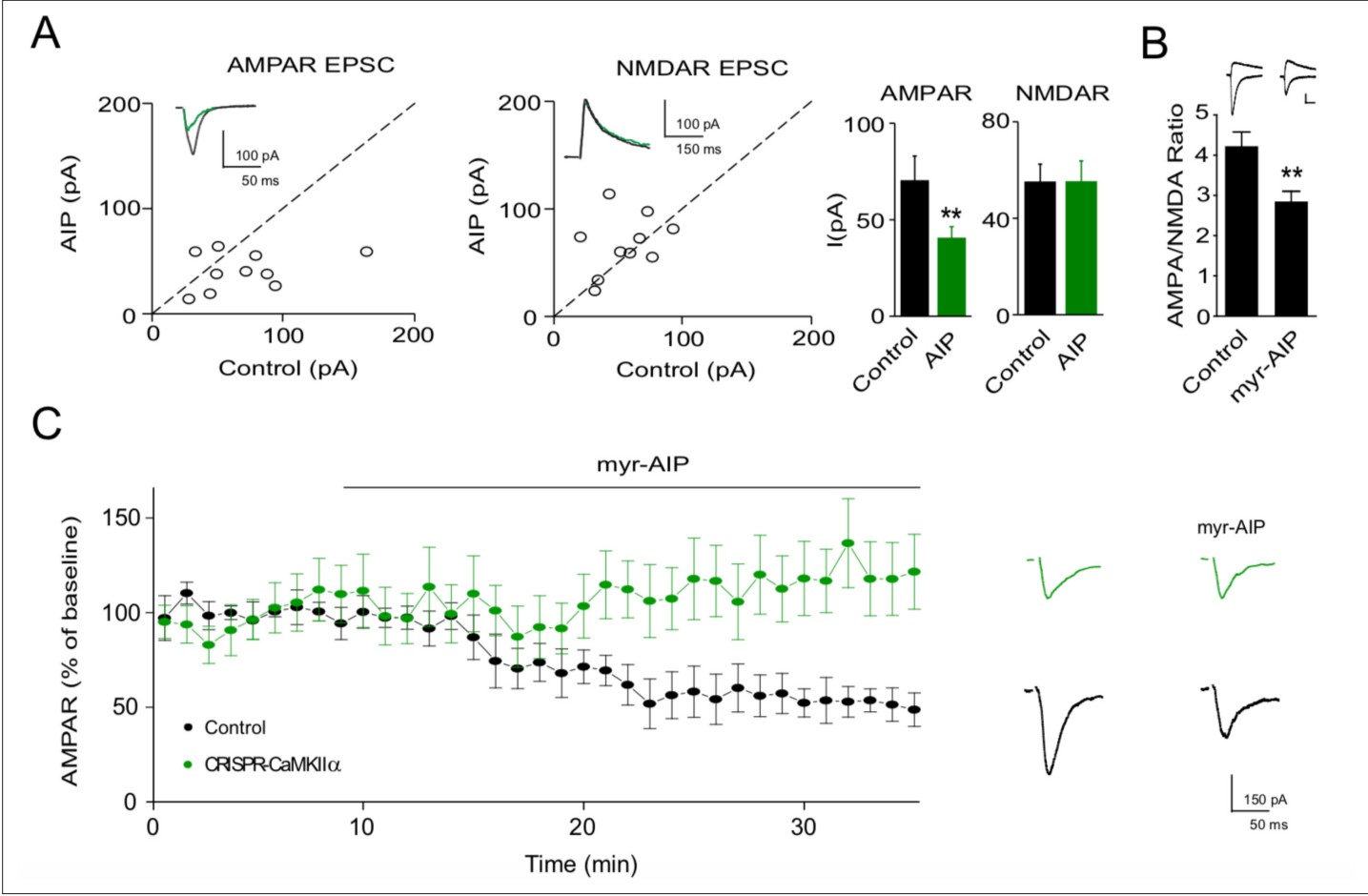

**Figure 2.** Ca²⁺-calmodulin-dependent kinase II (CaMKII) inhibitor AIP inhibits AMPAR synaptic transmission through CaMKIIα. (**A**) In culture slices, transfection of AIP reduced AMPAR EPSCs (left, *n* = 10, p < 0.01, two-tailed Wilcoxon Signed Rank Test) but not NMDAR EPSCs (right, *n* = 10, p > 0.05, two-tailed Wilcoxon Signed Rank Test). Sample traces show the effects of AIP on AMPAR EPSCs and NMDAR EPSCs. (**B**) AMPA/NMDA ratios compared to wild-type (WT) (*n* = 34) are reduced after myr-AIP (20 μM) treatment (*n* = 12). Scale bar = 50 pA vertical and 30 ms horizontal. (**C**) Left, summary of the effect of myr-AIP (20 μM) on AMPAR EPSCs in wt cells (*n* = 6) and interleaved CRISPR-CaMKIIα transfected cells (*n* = 5) from culture slices, normalized to each cell's baseline. The difference between control and CRISPR-CaMKIIα transfected cells at 30 min: p < 0.01, Mann–Whitney *U* test. Right, sample traces showing myr-AIP inhibition of AMPAR EPSC in control cells, but not in CRISPR-CaMKIIα transfected cells. Black traces are control cell, green are transfected cell. Mean ± standard error of the mean (SEM).

The online version of this article includes the following source data for figure 2:

**Source data 1.** Ca²⁺-calmodulin-dependent kinase II (CaMKII) inhibitor AIP inhibits AMPAR synaptic transmission through CaMKIIα.

NMDAR activity as a source for the constitutive CaMKII activity. In addition, acute application of the NMDAR antagonist APV has little effect on AMPAR EPSCs (*Incontro et al., 2018*).

The constitutive activity might reflect an LTP process induced by slice preparation, which is known to depolarize cells and release glutamate. To address this possibility, we pretreated animals with a high dose of MK-801, which is known to block in vivo NMDAR responses (*Davies et al., 1988*). To verify that NMDARs were blocked by this procedure, we recorded NMDAR EPSCs from slices from these animals and found that the NMDAR EPSC was blocked (*Figure 4—figure supplement 1B*). Having confirmed the absence of NMDAR responses the slices were perfused with MK-801. Under such conditions myr-CN27 still exerted its inhibitory effect (*Figure 4—figure supplement 1C*).

Second, if CaMKII is involved in synaptic memory, one would expect that LTP acquired when the animal was alive, would contribute to synaptic transmission, as previously proposed (*Lisman, 2017*; *Sanhueza and Lisman, 2013*). If this is the case, there are a number of predictions. One would expect that the transient inhibition of CaMKII should result in long-lasting inhibition, akin to the resetting of a molecular switch. In a number of cells, we monitored the depression over time following the removal

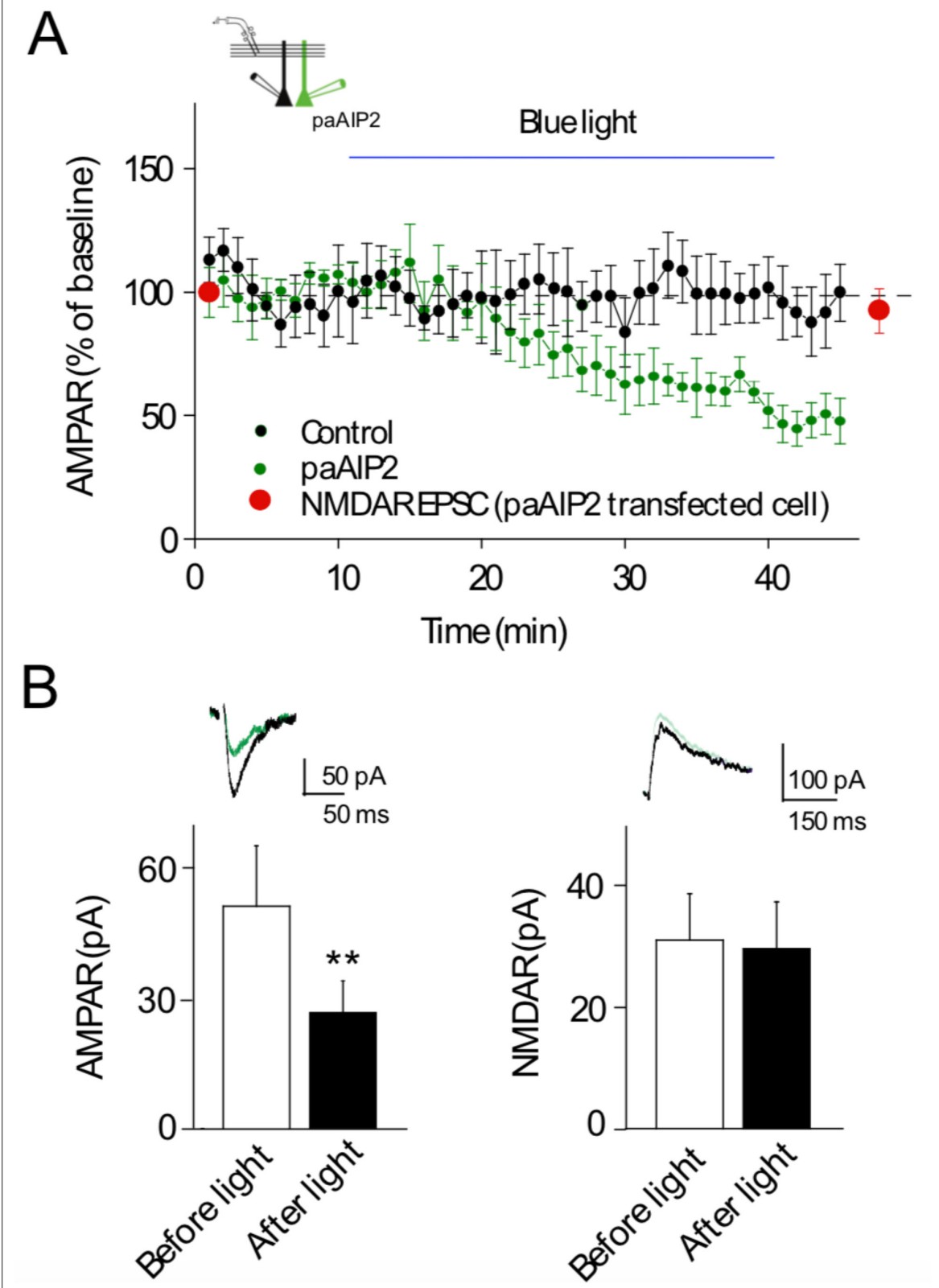

**Figure 3.** Light-activated Ca²⁺-calmodulin-dependent kinase II (CaMKII) inhibitor paAIP2 inhibits AMPAR synaptic transmission through CaMKIIα. (**A**) Time course of paAIP2 effect on AMPAR and NMDAR synaptic transmission in culture slices. Blue light exposure inhibited AMPAR synaptic transmission in paAIP2 expressing cells (green circles), but not in simultaneously recorded control cells (black circles) or NMDAR synaptic transmission (red circles). (**B**) Summary data showing that paAIP2 inhibited AMPAR synaptic transmission (before light: 51 ± 13.8 pA; after light: 26.5 ± 7.2 pA; $n$ = 12, p < 0.01, two-

*Figure 3 continued on next page*

*Figure 3 continued*

tailed Wilcoxon Signed Rank Test), but had no effect on NMDAR synaptic transmission (before light: 31.2 ± 7.6 pA; after light: 29.8 ± 7.7 pA; *n* = 5, p > 0.05, two-tailed Wilcoxon Signed Rank Test). Mean ± standard error of the mean (SEM). **p < 0.01.

The online version of this article includes the following source data and figure supplement(s) for figure 3:

**Source data 1.** Light-activated Ca$^{2+}$-calmodulin-dependent kinase II (CaMKII) inhibitor paAIP2 inhibits AMPAR synaptic transmission through CaMKIIα.

**Figure supplement 1.** Characterization of synaptic transmission in control cells and paAIP2 transfected cells from culture slices, before and after blue light exposure.

**Figure supplement 2.** CRISPR deletion of Ca$^{2+}$-calmodulin-dependent kinase II (CaMKIIα) prevents paAIP2 effect on AMPAR synaptic transmission in culture slices.

of myr-CN27 and found no recovery (*Figure 4A*). One might argue that myr-CN27 fails to washout of the cells. To circumvent this possibility, we repeated these experiments with paAIP2 whose action rapidly reverses upon terminating the blue light (~40 s). Again, we failed to observe any recovery (*Figure 4B*). We confirmed that this was not due to the continued action of the inhibitor, by demonstrating that LTP could be induced following light termination (see below, *Figure 5*).

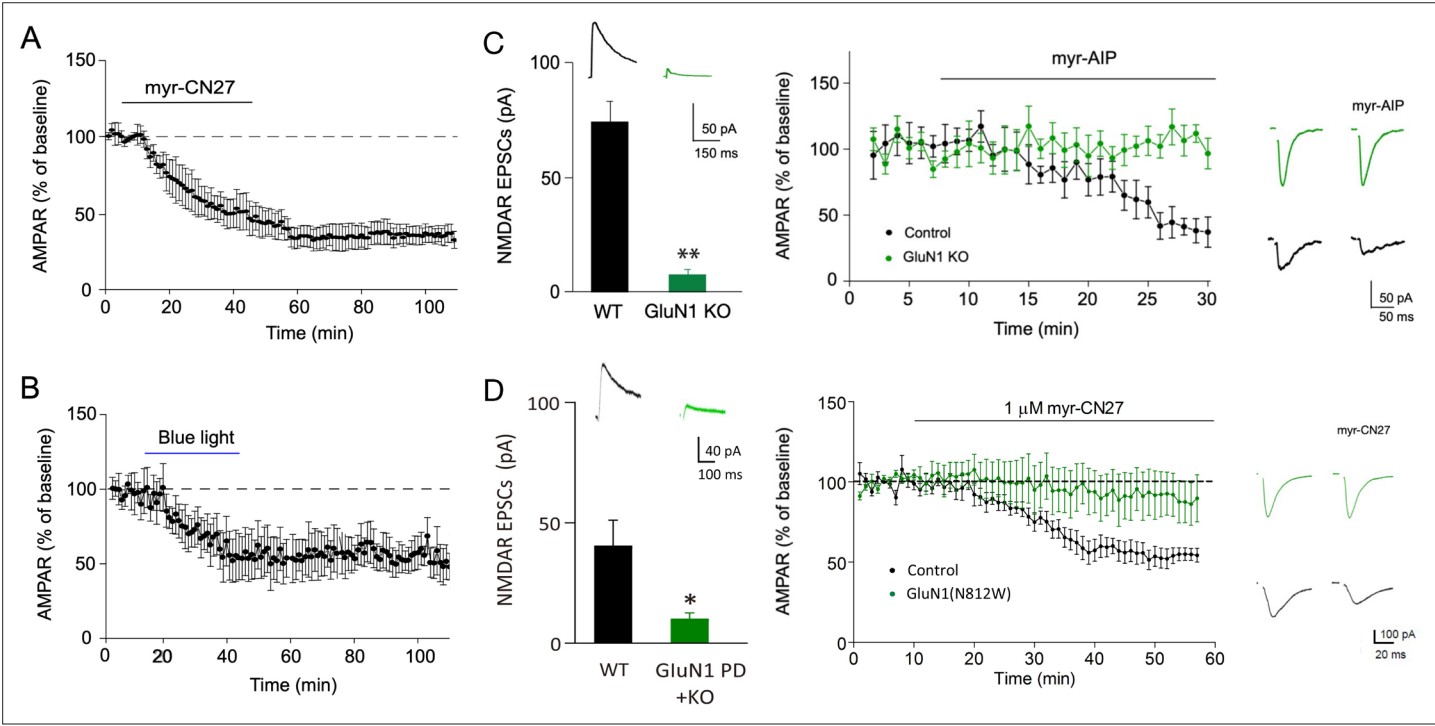

**Figure 4.** The depression caused by transient inhibition of Ca$^{2+}$-calmodulin-dependent kinase II (CaMKII) is long lasting and is absent in cells lacking NMDARs. (**A**) In acute slices, time course showing that following myr-CN27 application AMPAR synaptic transmission remains depressed (the difference between before and after myr-CN27: *n* = 4, p < 0.01, two-tailed Wilcoxon Signed Rank Test). (**B**) In culture slices, time course showing that following blue light exposure on paAIP2 expressing cells AMPAR synaptic transmission remains depressed (the difference between before and after blue light exposure: *n* = 5, p < 0.01, two-tailed Wilcoxon Signed Rank Test). (**C**) Left panel, paired recordings from acute slices of control cells and those expressing Cre in GluN1 floxed mice (*n* = 5, p < 0.01, two-tailed Wilcoxon Signed Rank Test). Middle panel, time course of the myr-AIP effect on AMPAR synaptic transmission in control cells (*n* = 5) and interleaved cells expressing Cre in GluN1 floxed mice (*n* = 5). p < 0.01, Mann–Whitney *U* test. Right panel, sample traces showing myr-AIP inhibited AMPAR synaptic transmission in control cell (black traces), but not in a GluN1KO cell (green traces). Mean ± standard error of the mean (SEM). (**D**) Left panel, paired recordings from acute slices of control cells and those expressing Cre and N812W (pore dead, PD) in GluN1 floxed mice (*n* = 6, p < 0.05, two-tailed Wilcoxon Signed Rank Test). Middle panel, the action of myr-CN27 is not rescued by expressing a GluN1 pore dead mutant (*n* = 8). Right panel, sample traces showing myr-CN27 inhibited AMPAR synaptic transmission in control cell (black traces), but not in a GluN1KO + PD cell (green traces).

The online version of this article includes the following source data and figure supplement(s) for figure 4:

**Source data 1.** The depression caused by transient inhibition of Ca$^{2+}$-calmodulin-dependent kinase II (CaMKII) is long lasting and is absent in cells lacking NMDARs.

**Figure supplement 1.** Origin of constitutive Ca$^{2+}$-calmodulin-dependent kinase II (CaMKII) effect on synaptic transmission in acute slices.

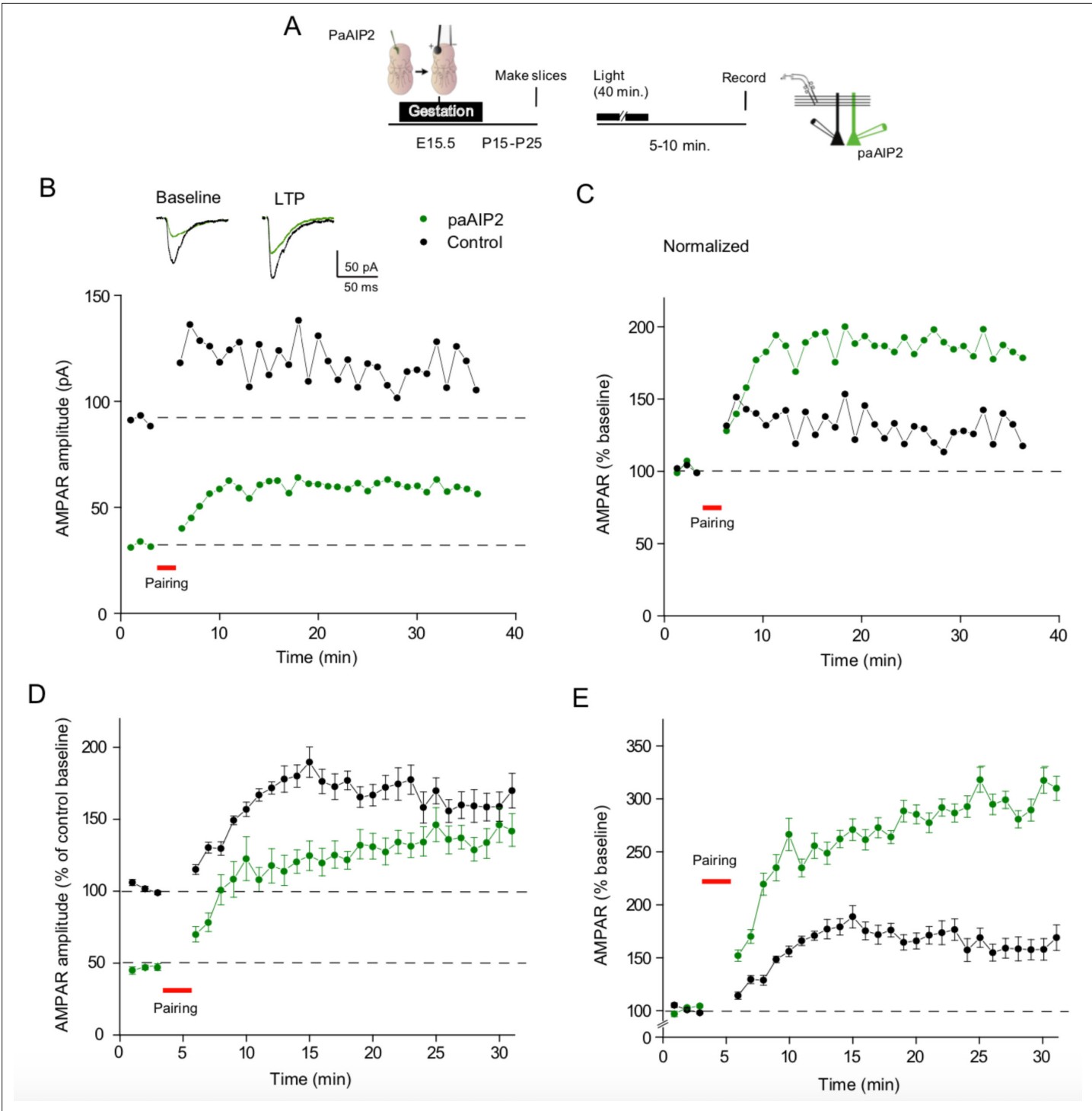

**Figure 5.** Transient inhibition of $Ca^{2+}$-calmodulin-dependent kinase II (CaMKII) enhances subsequent long-term potentiation (LTP). (**A**) Timeline of experimental procedure. Following in utero electroporation acute slices were prepared at P15–P25. Slices were then exposed to blue light for 40 min. Paired recordings were then made from a control cell (black circles) and a paAIP2 expressing cell (green circles). (**B**) One representative experiment showing that prior paAIP2 activation reduced baseline AMPAR EPSCs, but LTP was larger than that recorded in the neighboring control cell. (**C**) Normalized representative experiment showing that prior paAIP2 activation increased LTP. (**D**) Summary data showing that prior paAIP2 activation reduced baseline AMPAR EPSCs, but enhanced LTP (n = 5). (**E**) Normalized summary data showing that prior paAIP2 activation doubles the size of LTP. n = 5, p < 0.01, two-tailed Wilcoxon Signed Rank Test. Mean ± standard error of the mean (SEM).

The online version of this article includes the following source data for figure 5:

**Source data 1.** Transient inhibition of $Ca^{2+}$-calmodulin-dependent kinase II (CaMKII) enhances subsequent long-term potentiation (LTP).

If the constitutive activity resulted from prior NMDAR-dependent LTP, one would expect that this inhibitory action should be absent in cells in which NMDARs have been deleted embryonically. We carried out in utero electroporation to express cre recombinase in neurons in GluN1 floxed mice. Transfected cells were recorded at P16–20 in acute slices. We first carried out paired recordings in the AMPAR inhibitor CNQX to ensure that cre expressing neurons lacked NMDAR EPSCs (*Figure 4C*, left panel). We then compared the action of myr-AIP in transfected cells with that in interleaved control cells and normalized the baseline to 100%. Myr-CN27 had no effect in these cells. Application of myr-AIP had no effect in these cells (*Figure 4C*, middle and right panels, green circles and traces), whereas it had its normal inhibitory effect in interleaved control cells (*Figure 4C*, black circles and traces), further supporting the conclusion that the persistent CaMKII activity represents prior LTP. A caveat to these experiments is that the synaptic action of CaMKII requires its binding to the GluN2B C-terminus of the NMDAR. Thus, the loss of GluN2B in the GluN1 lacking neurons could equally explain the loss of effect of myr-CN27. To address this concern we repeated these experiments, replacing GluN1 with a pore dead GluN1 mutant (N812W) (*Amin et al., 2017*). While this mutant assembles normally with GluN2 subunits and traffics normally to the membrane, it generates little current (*Amin et al., 2017*). We carried out in utero electroporation in GluN1 floxed mice to transfect Cre and GluN1(N812W). Thus, this genetic experiment is the equivalent of having APV present in utero and until acute slices were made at P14–18. The NMDAR EPSC in transfected neurons was severely reduced, when compared to neighboring control cells recorded in CNQX (*Figure 4D*, left panel). The depressant effect of myr-CN27 recorded in these transfected cells was absent (*Figure 4D*, middle and right panels). These results suggest that in behaving animals, cells that lack NMDAR function, but with GluN2B intact, $Ca^{2+}$ signaling independent of the NMDAR appears incapable of activating CaMKII, at least in regard to its action on synaptic function.

Since LTP is saturable, one might expect that when constitutive activity is *transiently* silenced, the magnitude of LTP should be larger than that in control conditions. To test this prediction, we transfected paAIP2 in utero and prepared acute slices at days P15–25. The slices were exposed to blue light for 40 min and then simultaneous recordings were made from a transfected cell and a control cell (*Figure 5A*). In a typical experiment (*Figure 5B*), as well as in the summary of all experiments (*Figure 5D*), the size of baseline EPSCs in paAIP2 expressing cells following blue light exposure was, as expected, approximately half of that of the control neighboring cells. LTP was then induced in both cells. To compare the magnitude of LTP in the two cells we normalized the baseline EPSCs to 100%. As is clear, both in the single experiment (*Figure 5C*) and in the summary (*Figure 5E*), the magnitude of LTP in cells expressing paAIP2 is approximately twice that of control cells. This interaction between the constitutive potentiation and LTP is consistent with their sharing the same underlying mechanism. Taken together these results strongly support the conclusion that while the animal is alive, LTP is acquired and leaves a synaptic memory trace.

## Blocking CaMKII erases LTP

The experiments presented thus far are all consistent with the notion that synapses acquire LTP that contributes to synaptic transmission. If this is correct, then inhibiting CaMKII *after* LTP induction must reverse the potentiation. Yet, as discussed in the introduction, this experiment has failed on numerous occasions (*Buard et al., 2010*; *Chen et al., 2001*; *Malinow et al., 1989*; *Murakoshi et al., 2017*; *Otmakhov et al., 1997*), but see *Feng, 1995*. Thus, it was mandatory that we revisit this crucial experiment. We therefore carried out a 'two pathway experiment', in which the responses from two independent pathways were recorded from a single cell (*Figure 6A*). Following the recording of baseline responses, LTP was induced in one of the pathways (*Figure 6B*, red circles) by pairing depolarization of the cell to 0 mV with continued synaptic stimulation, while in the other control pathway, stimulation was stopped during the depolarization (*Figure 6B*, black circles). Once LTP had been established, myr-CN27 was applied and the effect on both pathways recorded. Myr-CN27 caused a complete reversal of established LTP and as expected, a ~50% reduction in the control pathway (*Figure 6B*). A summary of all the experiments (*Figure 6B*, n = 11) demonstrates that in the presence of myr-CN27 the LTP pathway and control pathway converge to approximately 50% of the control. This experiment establishes the essential role of CaMKII in maintaining LTP. A previous field potential study using tatCN21 (5 µM) failed to observe a reversal of LTP (*Buard et al., 2010*). We, therefore, repeated our two pathway experiment using field potential recording. Again, we observed that myr-CN27 (1 µM)

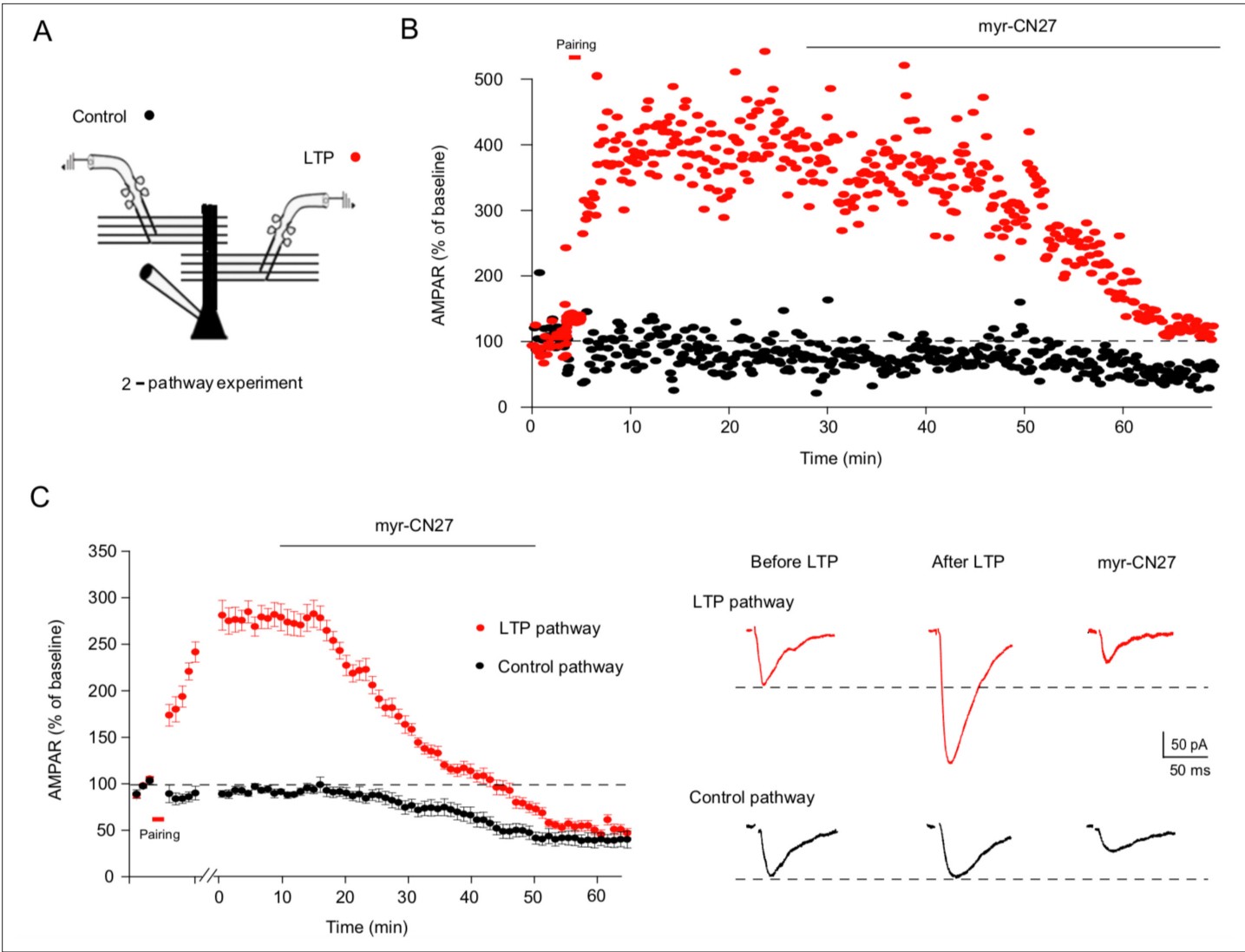

**Figure 6.** Myr-CN27 erases long-term potentiation (LTP) in acute slices. (**A**) Cartoon diagram of two pathway experiment. To record the response from two independent pathways, two bipolar stimulating electrodes were positioned to either side of the recorded cell with a distance of around 100 μm. Stimuli were applied alternately every 20 s. (**B**) A sample experiment showing that myr-CN27 inhibited the LTP pathway more strongly than the control pathway. (**C**) Left, summary data showing that myr-CN27 reduced the control pathway (black circles) 50%, while completely reversing LTP (red circles) (the difference between control and LTP pathway at 60 min: $n$ = 11, p > 0.05, two-tailed Wilcoxon Signed Rank Test). Responses are mean ± standard error of the mean (SEM). Right, sample traces showing the effect of myr-CN27 on AMPA EPSCs in control and LTP pathway. LTP is induced by 2 Hz stimulation for 90 s, while holding the cell at 0 mV.

The online version of this article includes the following source data and figure supplement(s) for figure 6:

**Source data 1.** Myr-CN27 erases long-term potentiation (LTP) in acute slices.

**Figure supplement 1.** Reversal of long-term potentiation (LTP) and reduction of synaptic transmission with application of 1 μM myr-CN27.

caused a slow reduction the EPSPs in the control pathway and a reversal of LTP (*Figure 6—figure supplement 1*). Give that tatCN21 (5 μM) had no effect on synaptic transmission (*Buard et al., 2010*; *Figure 1—figure supplement 1*), it is not surprising that it failed to reverse LTP in the previous study (*Buard et al., 2010*).

## Discussion

The role of CaMKII in the maintenance of LTP and information storage has been one of the most vexing issues in the field of synaptic plasticity. On the one hand, the biochemical properties of CaMKII

have for decades made this molecule an extremely attractive candidate for molecular storage (*Bhattacharyya et al., 2020*; *Coultrap and Bayer, 2012*; *Hell, 2014*; *Lisman et al., 2002*; *Lisman et al., 2012*). On the other hand, numerous physiological experiments over the years have failed to support the role of CaMKII in the *maintenance* of LTP and by extension memory. To reevaluate the physiological role of CaMKII in synaptic memory, we have used two classes of CaMKII inhibitory peptides. With these inhibitors we find that CaMKII is required for the maintenance of LTP and provide evidence that LTP acquired while the animal was alive leaves a lasting synaptic memory trace.

The first class of inhibitory peptide we used is derived from an endogenous CaMKII inhibitory protein referred to as CaMKIINtide (CN27) (*Chang et al., 1998*; *Goold and Nicoll, 2010*; *Pellicena and Schulman, 2014*; *Vest et al., 2007*). The second class of inhibitory peptide is derived from the autoinhibitory domain of CaMKII (*Bayer et al., 2001*; *Leonard et al., 1999*). Although these inhibitory peptides were thought to bind to separate sites (S and T sites), recent structural studies of the binding of GluN2B and other interacting peptides to CaMKII indicate that these peptides use similar interactions to bind across the substrate binding pocket of the CaMKII active site (*Özden et al., 2020*). Thus, at present it is not possible to use these two classes of inhibitors to distinguish between the disruption of GluN2B binding and the blockade of kinase activity. These peptides included a myristoylated version of CN27 (myr-CN27), AIP (myr-AIP) and a recently developed photoactivatable peptide inhibitor (paAIP2) (*Murakoshi et al., 2017*). Both classes of peptides block kinase activity with high affinity (*Chang et al., 2001*; *Ishida et al., 1998*), but also interfere with the binding of CaMKII to GluN2B (*Sanhueza et al., 2011*; *Vest et al., 2007*).

Our experiments clearly establish that inhibitory peptides fully reverse LTP. How long is CaMKII responsible for maintaining LTP? Experiments addressing the role of constitutive CaMKII in the maintenance of LTP are constrained by the length of recording; approximately 1 hr for whole cell and a few hours for field potentials. However, if LTP is the substrate for memory, it should leave a lasting memory trace at synapses, acquired while the animal is alive (*Lisman, 2017*; *Sanhueza and Lisman, 2013*). A number of studies have reported a lasting depression following transient inhibition of CaMKII (*Barcomb et al., 2016*; *Gouet et al., 2012*; *Sanhueza et al., 2011*; *Sanhueza et al., 2007*). Furthermore, even under basal conditions, CaMKII is found in isolated PSDs (*Petersen et al., 2003*; *Strack et al., 1997*) and synaptic puncta (*Bayer et al., 2006*) and this is in the autophosphorylated state (*Strack et al., 2002*; *Trinidad et al., 2005*). As discussed above all three inhibitory peptides (myr-CN27, myr-AIP, and paAIP2) depress synaptic transmission and this depression requires the presence of CaMKII. A number of trivial explanations, such as ongoing constitutive $Ca^{2+}$ activation of the enzyme, were excluded.

We then focused on the possibility that this constitutive action of CaMKII reflects prior LTP. First, the finding that the inhibition does not affect NMDAR responses is consistent with an LTP mechanism, since LTP preferentially enhances AMPAR responses (reviewed in *Nicoll, 2017*). Second, there should be no recovery from the transient inhibition of CaMKII, analogous to the resetting of a molecular switch. This is, indeed, the case. Third, cells in which the NMDAR has been deleted embryonically should be devoid of the constitutive activity. Again, this is the case. This is unlikely to be due to the loss of GluN2B subunit, because cells expressing a pore dead NMDAR mutant, in which assembles with GluN2B, had little CaMKII constitutive activity. This finding suggests that non-NMDAR sources of $Ca^{2+}$ in the behaving animal are incapable of triggering CaMKII-dependent enhancement in synaptic transmission. This tight linking of NMDAR sources of $Ca^{2+}$ to CaMKII is critical, because it prevents the degrading of the Hebbian nature of plasticity. Finally, it is well established that LTP is saturable. Thus, if constitutively action CaMKII represents LTP, removing this component should allow for larger LTP. We find that in cells in which CaMKII is transiently blocked LTP is approximately twice as large as in neighboring control cells. Taken together these results suggest that constitutive CaMKII represents LTP acquired at synapses while the animal was alive, thus supporting a role for CaMKII in synaptic memory.

Based on the present results the term 'basal synaptic transmission' needs to be reevaluated. It has generally been assumed that the synapses studied in a hippocampal slice, in which much of the afferent drive from multiple inputs have been removed in the slicing, are at a ground 'basal' state. The present results indicate that the synaptic currents we measure are actually maintained by a persistent enhancement acquired prior to slicing.

It should be pointed out that the contribution of CaMKII to synaptic transmission is unlikely to be a simple addition to the overall preexisting excitatory drive onto the cell. It has long been argued that

such a scenario would be unstable and quickly saturate (*Bienenstock et al., 1982*; *Cooper and Bear, 2012*; *Fusi et al., 2005*; *Morrison et al., 2008*; *Toyoizumi et al., 2014*; *Turrigiano, 2008*). A number of well-established mechanisms exist to maintain stability, including heterosynaptic depression (*Scanziani et al., 1996*), NMDAR-dependent long-term depression (*Malenka and Bear, 2004*) and on a longer time scale cell wide homeostasis (*Turrigiano, 2008*). Thus, it is envisaged that the CaMKII memory trace is embedded in a network of synapses with no overall net change in the excitatory drive onto the cell or network.

It is striking to compare the action of paAIP2 on LTP induction (*Murakoshi et al., 2017*) and on synaptic transmission (present study). A brief light exposure (~1 min), delivered immediately prior to inducing LTP, is sufficient to prevent LTP (*Murakoshi et al., 2017*). This finding indicates that the blocking of CaMKII activation by paAIP2 is very rapid. In contrast, the effect on synaptic transmission and the reversal of LTP required 10's of minutes. What might explain the dramatic difference in the kinetics of inhibition of LTP induction compared to LTP maintenance and constitutive CaMKII? There is a well-accepted sequence of events for the role of CaMKII in the induction of LTP (*Coultrap and Bayer, 2012*; *Hell, 2014*; *Lisman et al., 2012*; *Nicoll, 2017*). Following NMDAR activation $Ca^{2+}$ binds to CaM, which then binds to CaMKII activating the enzyme. This causes the autophosphorylation of T286 and the translocation of cytosolic CaMKII to the PSD where it binds to the C-tail of GluN2B. It seems reasonable to expect that under baseline conditions much of CaMKII and paAIP2 are freely diffusible in the spine cytoplasm, as are phosphatases. The activation of paAIP2 would quickly inhibit CaMKII autophosphorylation and its recruitment to the PSD and phosphatases would quickly reverse its effect. During baseline transmission and during the maintenance of LTP a fraction of CaMKII is bound to GluN2B. Evidence suggests that the CaMKII/GluN2B complex is protected from phosphatases (*Cheriyan et al., 2011*; *Lisman and Raghavachari, 2015*; *Mullasseril et al., 2007*). Thus, reversing action of this sequestered CaMKII would be more difficult than preventing phosphorylation of cytosolic CaMKII.

The present results could shed light on the underlying mechanism of long-term depression (LTD). Although this topic remains contentious (*Collingridge et al., 2010*; *Dore et al., 2016*; *Goodell et al., 2017*; *Lisman and Zhabotinsky, 2001*; *Malenka and Bear, 2004*; *Stein et al., 2021*; *Wong and Gray, 2018*), the long-held view is that LTD is mediated by the activation of phosphatases, in particular protein phosphatase 1 (PP1) (*Mulkey et al., 1994*; *Mulkey et al., 1993*). One of the limitations to this model is the lack of evidence that synaptic transmission is maintained by kinase activity. Our demonstration that CaMKII maintains synaptic transmission provides a very simple model for LTD, that is, a depotentiation or reversal of this prior LTP. One of the features of LTD is that it takes many minutes for its induction. Our finding that it takes many minutes for CaMKII inhibitors to depress synaptic transmission provides an explanation for this property. A formal model for the interplay between CaMKII and PP1 was proposed some time ago (*Lisman and Zhabotinsky, 2001*). The absence of LTD in CaMKII knockout mice (*Coultrap et al., 2014*; *Stevens et al., 1994*) supports such a model, but does not rule out other models.

What might account for the failure of previous studies to detect a component of synaptic transmission driven by constitutive CaMKII (*Achterberg et al., 2014*; *Buard et al., 2010*; *Chen et al., 2001*; *Feng, 1995*; *Giese et al., 1998*; *Malinow et al., 1989*; *Murakoshi et al., 2017*; *Otmakhov et al., 1997*; *Silva et al., 1992*; *Wang and Kelly, 1996*), but see *Hinds et al., 1998*, or to detect a role of constitutive CaMKII in maintaining LTP (*Buard et al., 2010*; *Chen et al., 2001*; *Malinow et al., 1989*; *Murakoshi et al., 2017*; *Otmakhov et al., 1997*), but see *Feng, 1995*? We suggest three factors. The first concerns the duration of peptide application, which in our hands takes 10's of minutes to act, especially with field potential recording. The second concerns the concentration and time. As discussed above it is more difficult to reverse the constitutive action of CaMKII, than it is to block its activation. Finally, the membrane permeabilizing agent is important. In our hands myristoylation was more effective and specific than the CPP, tat.

In summary our results overcome many of the obstacles that have prevented embracing CaMKII as a molecular storage device. Specifically, our results show that transient inhibition of CaMKII results in a lasting depression of synaptic transmission with properties consistent with the erasure of prior LTP acquired while the animal was alive. Furthermore, the finding that CaMKII inhibition *after* the induction of LTP reverses LTP establishes its role in LTP maintenance. Thus, these physiological results compliment the rich biochemical literature on CaMKII, making CaMKII a particularly attractive molecular

storage device. It is important to note, that our results have focused on LTP and it remains open as to whether CaMKII actually stores memories. Recent experiments (*Rossetti et al., 2017*) using the viral expression of the catalytically dead CaMKII K42M mutant, presumed to act in a dominant negative manner, support its role in memory.

## Materials and methods

### Animals

All the experimental procedures on animals were approved by the UCSF Animal Care and Use Committee, BUA # BU002466-04C. For acute slice recordings, typically we use 5–10 animals to obtain complete dataset; for culture slice recordings, we usually used 5–7 animals to obtain complete dataset.

### Experimental constructs and chemical agents

The paAIP2 plasmid was obtained from Dr. Ryohei Yasuda and had been characterized (*Murakoshi et al., 2017*). The CRISPR construct targeting at CaMKIIα was previously characterized (*Incontro et al., 2018*). For biolistic experiments, all the plasmids were expressed in pCAGGS vector, which contains an internal ribosome entry site (IRES) followed by the fluorophore GFP. myr-CaMKIINtide (myr-CN27) and myr-AIP were purchased from Calbiochem. Inc (catalog#, 208921; catalog#, 189482). MK-801 maleate was purchased from HelloBio. Inc (catalog#, HB30004). Myr-CN21 was custom ordered from Elim Biopharm, Inc by myristoylating the N-terminal of CN21:

KRPPKLGQIGRSKRVVIEDDR amino acid sequence.

### Slice culture and biolistic transfection

Hippocampal cultured slices are obtained from 6- to 8-day-old rats (*Stoppini et al., 1991*). Biolistic transfection was done 1 day after sectioning, by using a Helio Gene Gun with 1 µm DNA-coated gold particles (BioRad). Slices were maintained at 34°C and the medium was changed every 2 days. Typically, slices are used for electrophysiological recording 6–8 days after transfection, except for CRISPR experiments, in which slices are maintained for another week before recording.

### In utero electroporation

E15.5 pregnant mice were anesthetized with 2.5% isoflurane in $O_2$ and injected with buprenorphine for analgesia. The lateral ventricles of embryos were injected with 1 µl mix plasmid DNA (1 µg/µl) with beveled micropipette. Each embryo was electoporated with 5 × 50 ms, 35–40 V pulse.

### Acute slice preparation

Mice of 2–3 weeks of age were anesthetized with 4% isoflurane, decapitated, and the brain dissected free. The whole brain was sliced into 300 µm slices in cutting solution as described (*Granger et al., 2013*); recovery at 34°C for half an hour and then stored at room temperature. Solutions were continuously gassed with 95% O2/5% CO2.

### Photostimulation

Blue light pulses (0.1 Hz, 1 s, 20 mW/cm$^2$) were generated by 473 nm blue DPSS laser (Shanghai Laser & Optics Century, BL473T8-300FC). The blue light from a laser was delivered through a optical patch cable connected to the optical fibers. Light pulses were controlled by a Master-8 (A. M. P. I). The blue light is applied for 1-s duration with interval of 10 s.

### Injection of MK-801

MK-801 was dissolved in saline solution and administered as a single injection i.p. (10 mg/kg body weight) 1 hr before brain slicing.

### Electrophysiological recording

Whole-cell voltage clamp recordings were obtained from either wild-type cells or fluorescent transfected pyramidal cell in CA1 region of hippocampus (*Schnell et al., 2002*). Where indicated dual recordings were made from control and transfected cells. Pyramidal neurons were identified by location and morphology. All recordings were made at 20–25°C. Internal solution (in mM): 135 CsMeSO$_4$,

8 NaCl, 10 HEPES, 5 QX314-Cl, 4 Mg-ATP, 0.3 Na-GTP, 0.3 EGTA, 0.1 spermine. Osmolarity was adjusted to 290–295 mOm and pH was buffered at 7.3–7.4. External solution (mM): 119 NaCl, 2.5 KCl, 4 $CaCl_2$, 4 $MgCl_2$, 1 $NaH_2PO_4$, 26.2 $NaHCO_3$, 11 glucose, bubbled continuously with 95% $O_2$/5% $CO_2$. For field recordings, the internal solution was 3 M NaCl with a large opening pipette tip. Synaptic currents were evoked every 10 s with bipolar stimulating electrodes placed in s. radiatum. To record EPSCs, picrotoxin (100 µM) was added to the external solution; for recording of AMPAR EPSCs, the cell membrane was held at −70 mV, while for NMDAR EPSCs, it was hold at +40 mV; LTP is induced by stimulating at 2 Hz for 90 s while clamping the cell at 0 mV. For two independent synaptic pathway experiments, two bipolar stimulating electrodes were positioned to each side of the recording electrode with a distance of around 100 µm and alternately stimulated every 20 s. Current responses were collected with a Multiclamp 700B amplifier (Axon Instruments), filtered at 2 kHz, and digitized at 10 kHz. Cells with series resistance larger than 20 MOhm were excluded from analysis.

## Statistical analysis

Data analysis was carried out in Igor Pro (Wavemetrics), Excel (Microsoft), and GraphPad Prism (GraphPad Software). All paired recording data were analyzed statistically with a Wilcoxon Signed Rank Test for paired data. For unpaired data, a Mann–Whitney *U* test was used. Statistical parameters including the types of the statistical tests used, exact value of *n*, precision measures (mean ± standard error of the mean) and statistical significance are reported in the figure legends. All statistical tests performed were two sided, and with all tests a p value of <0.05 was considered statistically significant. All error bars represent standard error of the mean.

## Acknowledgements

We wish to thank the late John Lisman and Johannes Hell for their unfailing input during the course of this study. We thank Dr. R Yasuda for the paAIP2 plasmids, Dr. Wei Lu for contributing the data for Figure 3A and Dr. Lonnie Wullmuth for GluN1 pore dead plasmid (GluN1(N812W)). We thank members of the Nicoll lab for comments on the manuscript. We appreciate the technical assistance from Eric Dang. This research was supported by grant MH070957.

## Additional information

### Funding

| Funder | Grant reference number | Author |
|---|---|---|
| National Institutes of Health | MH070957 | Roger A Nicoll |

The funders had no role in study design, data collection, and interpretation, or the decision to submit the work for publication.

### Author contributions

Wucheng Tao, Roger A Nicoll, Conceptualization, Data curation, Formal analysis, Funding acquisition, Investigation, Methodology, Project administration, Resources, Software, Supervision, Validation, Visualization, Writing – original draft, Writing – review and editing; Joel Lee, Investigation; Xiumin Chen, Data curation, Investigation, Validation; Javier Díaz-Alonso, Jing Zhou, Samuel Pleasure, Methodology

### Author ORCIDs

Wucheng Tao ⓘ http://orcid.org/0000-0003-2577-8161
Javier Díaz-Alonso ⓘ http://orcid.org/0000-0002-4980-7441
Jing Zhou ⓘ http://orcid.org/0000-0003-2809-7097
Samuel Pleasure ⓘ http://orcid.org/0000-0001-8599-1613
Roger A Nicoll ⓘ http://orcid.org/0000-0002-6977-4632

## Ethics

All the experimental procedures on animals were approved by the UCSF Animal Care and Use Committee, BUA # BU002466-04C.

## Decision letter and Author response

Decision letter https://doi.org/10.7554/eLife.60360.sa1
Author response https://doi.org/10.7554/eLife.60360.sa2

## Additional files

### Supplementary files

• Transparent reporting form

### Data availability

All data generated or analysed during this study are included in the manuscript and supporting files.

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
