## [Editor Report]

The article addresses the ‘decades-old’ unresolved question as to whether CaMKII is required for the maintenance of synaptic long-term potentiation, and shows, based on a set of elegant experiments, that it indeed is.

---

## [Decision Letter]

**Decision letter after peer review:**

Thank you for submitting your article "Synaptic memory is maintained by persistent CaMKII activity" for consideration by *eLife*. Your article has been reviewed by 3 peer reviewers, one of whom is a member of our Board of Reviewing Editors, and the evaluation has been overseen by John Huguenard as the Senior Editor. The reviewers have opted to remain anonymous.

The reviewers have discussed the reviews with one another and the Reviewing Editor has drafted this decision to help you prepare a revised submission.

Summary

Tao et al. describe a study on an important issue in the field of nerve cell biology that has been unresolved for decades despite substantial research efforts: The role of CaMKII in the maintenance of LTP. By using various means to perturb CaMKII activity and activation, the authors provide evidence for the notion that CaMKII remains activated for extended periods of time after LTP induction and might thus be part of a synaptic memory mechanism.

Essential revisions

Revisions Requiring Additional Data/Experiments

1. The new data of the present study are in disagreement with a host of previously published studies, which indicated that CaMKII activation is generally rather short-lived and transient. Instead, the authors propose that a subfraction of CaMKII remains activated to store memory. The authors' main explanation for the data discrepancy (i.e. prolonged inhibition >30 min) is not fully convincing as some effects documented in the present study are seen earlier and within a time frame that others have also looked at. The reviewers are missing experimental approaches to resolve the striking data discrepancy, e.g. by comparing their assay conditions with those used by others.

2. No data to identify the persistently active CaMKII subfraction or its target proteins are presented, leaving the entire notion of memory-CaMKII still speculative. The manuscript would be strengthened substantially with direct evidence on the 'memory-CaMKII', e.g. biochemical proof of activated CaMKII. For instance, the authors could test their model of GluN2B/CaMKII interaction biochemically with CoIPs to identify the constitutively active CaMKII (e.g. through T286 phosphorylation, which should be absent with no LTP/memory). This experiment could be controlled with a GluN2B mutant that lacks the interaction.

3. The control experiment for no LTP is problematic. Knocking out NMDA receptors increases synaptic strength, as the authors showed previously. Furthermore, NR2B, which the authors propose in their model to be the scaffolding protein for active CaMKII, is not present. Thus, multiple changes occur in this experiment, of which the authors pick one as support for their claim that LTP is missing. The authors need to show how this control prevents the formation of the 'memory-CaMKII'. It seems that this is not possible with the documented experiment as synaptic GluN2B is removed, which is proposed to be the candidate interactor/scaffold of the 'memory-CaMKII'.

4. Figure 4 and S4: The run-up of the baseline has a similar slope as the increase after LTP induction, which is problematic. Peptide experiments should be performed with fEPSP recordings with 'solid' baseline rather than 3-point 'baseline'. In essence, it would be helpful to have different LTP protocols to show the same effect. Using cell-permeable peptides does not require whole cell recordings. With fEPSP, the issue of LTP wash-out as a rationale for the short baseline can be circumvented.

5. Figure 6: Compared to Figure 4, the LTP in control is rather reduced and not enhanced in paAIP2. It is not uncommon that effect sizes vary, but here it is used as the single line of evidence to indicate that the peptides create more 'room' for later LTP. The experiment needs to be better controlled, e.g. with interleaved experiments.

Revisions Requiring Text Redaction

6. The peptides used to perturb CaMKII function are probably among the best tools currently available. However, their specificity remains an issue. They have never been tested systematically but only on a handful of targets. CaMKIINtides have other 'structural' effects than the one on NR2B binding, i.e. interactions with other proteins, and AIP peptides are known to have effects on processes that are not CaMKII-related (Pellicena and Schulman – Front Pharmacol 2014). This issue should be included in the discussion as a general caveat.

7. There are no data to show that LTP accumulates during life. Such evidence would be important to support the model propose. Age and LTP 'experience' are not generally applicable criteria when switching back and forth between cultured and acute slices. In essence, one would not expect 'memories' in slice culture, but the effects of the peptides appear to be the same in the two culture systems. With regard to the NMDAR KO experiment, it is then argued that it prevents the formation of in vivo memories and so that no effect of the peptide inhibitors is seen. Thus, whether in vivo or in vitro, synapses appear to 'learn' the same amount. The line of logic in the discussion of these data is not entirely convincing. Given that showing that the corresponding synapses have been LTPed in vivo is highly non-trivial, it is suggested that alternative models and explanations are discussed.

8. Generally, and in the context of the point above, the reviewers suggest to substantially tone down the claim that a mechanism of memory is identified. CaMKII has both basic and potentiation-related synaptic function, which all might well overlap.

9. LTP is synapse specific. Erasing LTP by blocking the memory molecule would thus be expected to be synapse specific as well. However, both control and LTP pathway are affected by CaMKII perturbation. The authors argue that this is due to LTP accumulation during life. The support for this claim is limited. As the authors show, basal synaptic transmission is reduced by the CaMKII inhibitors. Whether this is LTPed synaptic strength is not clear. An alternative explanation would be that during early LTP expression, before consolidation, LTP can be reversed by inhibiting CaMKII. This should be discussed.

10. In CaMKII-α KO mice, LTP is impaired but not absent (Silva et al. – Science 1992). Furthermore, CaMKII autophosphorylation was proposed to facilitate LTP induction, but is not essential (Chang et al. – Neuron 2017). These results are consistent with an important role of CaMKII in LTP, but indicate that the magnitude of LTP blockade depends on the induction protocol and is more in agreement with induction and consolidation of LTP rather than long-term memory. This should be discussed.

11. Halt et al. (EMBO J 2012) proposed that memory consolidation requires the CaMKII/NR2B interaction. The effects of the inhibiting peptides after 10s of minutes, as seen in the present study, are consistent with this notion; whether long-term memory is affected, as proposed in the present study, is not shown. The major novelty of the present study would be that a fraction of CaMKII remains constitutively active to store memory. Reversal of LTP in 30-60 min is not necessarily related to long-term memory, and the notion that reduction of basal transmission reflects the same mechanism is not straight forward. This should be discussed. If more evidence in support of the memory-CaMKII idea can be provided (see above), additional experiments in the context of this item are not expected.

12. Figure 2: The effect of AIP expression appears less than 50%. A summary point of two different cell populations is incorrect (see e.g. Levy et al. – Neuron 2015; Kim et al. – Neuron 2015). The data need to be plotted differently and an explanation as to why the effect is smaller needs to be attempted.

13. Figure S5: The effect of myr-CN27 with MK-801 appears smaller than in other conditions. Does activity in slices before recording have an effect?

14. Line 36: It is unclear what data would support the notion that LTP accumulates during life. The corresponding passage should be referenced or rephrased.

15. Lines 76-78: This claim is incorrect. Halt et al. proposed a function for CaMKII in memory consolidation.

[Editors' note: further revisions were suggested prior to acceptance, as described below.]

Thank you for resubmitting your work entitled "Synaptic memory is maintained by persistent CaMKII activity" for further consideration by *eLife*. Your revised article has been reviewed by 3 peer reviewers, one of whom is a member of our Board of Reviewing Editors, and the evaluation has been overseen by John Huguenard as the Senior Editor.

The manuscript has been improved very substantially, but there are several remaining issues that we regard to be important:

Data Interpretation and Discussion

1. CaMKII Enzymatic Activity vs. Protein Interactions. The firm conclusion that it is continued CaMKII activity rather than CaMKII binding to relevant postsynaptic binding partners (e.g. the C-terminus of GluN2B) is still not justified by the data. Notably, CaMKII binding to GluN2B was shown to be important for LTP (Barria and Malinow, 2005; Halt et al., 2012). Earlier evidence indicates that the CN27-related CN21 peptide reverses LTP by displacing CaMKII from GluN2B and not by inhibiting CaMKII but (Sanchueza et al., 2011). The authors argue that the AIP peptide only inhibits CaMKII activity without displacing it from GluN2B because earlier work showed that addition of untagged AIP during immunoprecipitation of CaMKII does not disrupt co-precipitation of the NMDAR at 20 μm (Leonard et al., 1999). However, although AIP does not disrupt the preformed CaMKII-NMDAR complex (Leonard et al., 1999), expression of it in HeLa cells prevents binding of CaMKII to GluN2B (see Suppl Figure 2 in Murakoshi et al., 2017). In other words, when expressed in intact cells, AIP does inhibit CaMKII binding to GluN2B. This could also be the case when myrAIP is applied to intact cells,. Perhaps in intact cells CaMKII has a tendency to unbind from GluN2B (potentially due to dephosphorylation of T286) and AIP augments this unbinding or prevents re-binding (potentially by inhibiting re-phosphorylation), while in vitro in immunoprecipitates there is no unbinding and AIP cannot displace CaMKII from GluN2B. The issue of AIP acting by inhibiting CaMKII activity vs. CaMKII binding to GluN2B is exacerbated by recently published data indicating that S and T site binding of substrates vs. docking proteins for CaMKII might not be separable (Özden et al., 2020: CaMKII binds both substrates 1 and effectors at the active site. bioRxivhttps://doi.org/10.1101/2020.10.25.354241). This work indicates that T and S sites form a continuum and that most – if not all – binding peptides will affect both, kinase activity and binding to anchoring proteins. Furthermore, the authors argue that their myrCN27 acts via CaMKII inhibition and not CaMKII displacement from GluN2B. Their reasoning is that they had a full effect at 1 μm of myrCN27 whereas Sanchueza et al. (2011) reported that 20 but not 5 μm of the related tat-CN21 displaced CaMKII from GluN2B (and that this correlated with a decrease in transmission and reversal of LTP during maintenance phase), when 5 μm tat-CN21 was sufficient to block CaMKII activity and LTP induction presumably because 5 μm is sufficient to block kinase activity. However, myrCN27 is longer than CN21 and the myr tag could promote binding of this peptide to CaMKII, so it could be much more potent than tat-CN21 in displacing CaMKII from GluN2B and already accomplish displacement at 1 uM.

Because of the considerations above, it is mandatory that the authors modify the respective discussion and do not dismiss the notion that 1 μm myrCN27 and AIP in general could have acted by displacing CaMKII.

2. In-Vivo-LTP. The authors argue that GluN1 KO prevents LTP expression in vivo and that this experimental approach could therefore test whether previous LTP in vivo is required for the dependence of synaptic strength on CaMKII. Indeed, the results are consistent with this line of argumentation. However, GluN1 KO has multiple additional effects: It increases synaptic strength and eliminates synaptic NR2B expression, which is thought to be the anchor for active CaMKII. Furthermore, multiple forms of synaptic plasticity (e.g. LTP, LTD, scaling, a.o.) occur during the lifetime of an animal, and notably, NMDARs, CaMKIIα, and CaMKIIαβ were reported to be involved in these forms of plasticity. Thus, the interpretation that in GluN1 KO neurons LTP is absent in vivo is reasonable, but given the other consequences of the KO, it is not conclusive. To further link the requirement of constitutively active CaMKII to in vivo LTP, the authors use a type of occlusion experiment. They express the photoactivatable AIP in neurons in utero. Then activate AIP for 40 min in slices after which they start recording. Like inhibition while stimulating, the inhibition without stimulation reduced AMPAR EPSC by 50% compared to neighboring control neurons. LTP in the previously inhibited neurons was about 2-times bigger than that in control neurons. The authors argue that the inhibition of CaMKII liberated "space" for more LTP. Given that LTP is saturable, the origin of this increased "space" is not clear. It could be due to a shift of the baseline further away from the ceiling or, as argued, to the reversal of in vivo LTP that is then newly triggered. Again, a suggestive interpretation, but not conclusive. In a final experiment, the authors report that 1 µM myr-CN27 reverts LTP during the expression phase. The experiment is performed in a two-pathway experiment, in which both the control non-induced pathway, as well as the induced pathway are reduced. Notably, the strength of the LTPed pathway settles at the same strength as the control pathway, indicating that both basal and the new LTPed synaptic strength is reduced. Myr-AIP is reported to have a similar effect at 20 µM, but not 1 µM. While the 1 µM serves as an important peptide control, the effect of the 20 µM is not as efficient as the 1 µM myr-CN27 as after around 60 min, some potentiation is left and EPSCs are not reduced to 50% of baseline. In a single experiment with field EPSP, myr-CN27 also reduces the fEPSPs of both the LTPed and control pathway, but in the time window observed, the responses do not converge. While these results indicate that CaMKII activity during expression is required, they do not necessarily translate into a mechanism selective for LTP, e.g. that the reduction in control pathway is based on in-vivo-acquired LTP. The different outcomes of the myr-CN27 peptide, the myr-AIP peptide in whole-cell recordings, and the myr-CN27 peptide in fEPSP are fully consistent with different mechanisms, e.g. with CaMKII activity being required for the expression of LTP, while structural assembly being required to stabilize basal synaptic transmission, as reported previously. To distinguish between these possibilities, information about on which CaMKII populations the different peptides operate would be important, e.g. whether myr-CN27 dissociates CaMKII from NR2B, or whether myr-AIP or myr-CN27 change the autoactivation of CaMKII.

In conclusion, the present study identifies an essential role of CaMKII activity for basal synaptic transmission and LTP expression. It is convincingly shown that this relates to differential effects of cell-penetrating peptides. However, the title and abstract appear too far-fetched as the results do not conclusively show that basal synaptic transmission is the product of acquired in vivo LTP. The reviewers urge the authors to address this issue by more conservative wording of title, abstract, and relevant text passages.

---

## [Author Response]

Essential revisionsRevisions Requiring Additional Data/Experiments1. The new data of the present study are in disagreement with a host of previously published studies, which indicated that CaMKII activation is generally rather short-lived and transient. Instead, the authors propose that a subfraction of CaMKII remains activated to store memory. The authors' main explanation for the data discrepancy (i.e. prolonged inhibition >30 min) is not fully convincing as some effects documented in the present study are seen earlier and within a time frame that others have also looked at. The reviewers are missing experimental approaches to resolve the striking data discrepancy, e.g. by comparing their assay conditions with those used by others.

This is a reasonable concern. We have rewritten the beginning of the Results section by discussing the background involved in our focusing on myr-CN27. We discuss the pros and cons of using cell permeabilizing peptides (CPPs) and protein lipidation, which we believe has an impact on our results. We also touch base with previous results with tatCN21. In addition, given that almost all of the previous data used field potential recordings, we repeated some of our key experiments with field potential recording. First, we tested myr-CN27 on baseline transmission. As shown in Author response image 1 (black symbols) the effect of myr-CN27 is considerably slower that seen with interleaved whole cell recording (blue symbols) and takes on average about 2 hours to stabilize. The most likely explanation for this is that with whole cell recording, cells near the surface are recorded, whereas with field potentials the peptide must diffuse throughout the slice.

**Author response image 1. sa2fig1:** 

Second, we repeated the myr-CN27 “eraser” experiment with field potential recording (Figure 6—figure supplement 1).

2. No data to identify the persistently active CaMKII subfraction or its target proteins are presented, leaving the entire notion of memory-CaMKII still speculative. The manuscript would be strengthened substantially with direct evidence on the 'memory-CaMKII', e.g. biochemical proof of activated CaMKII. For instance, the authors could test their model of GluN2B/CaMKII interaction biochemically with CoIPs to identify the constitutively active CaMKII (e.g. through T286 phosphorylation, which should be absent with no LTP/memory). This experiment could be controlled with a GluN2B mutant that lacks the interaction.

We believe that this type of experiment has already been reported. In the classic paper of Seth Grant {Husi, 2000 #1497} he showed the CoIP of CaMKII with NMDARs. Furthermore {Leonard, 1999 #2681} showed that CaMKII/NMDAR complex exists under basal conditions. Finally, Halt et al. {Halt, 2012 #2216} showed that this interaction is reduced in their knockin mouse, in which the C-tail of GluN2B no longer binds to CaMKII. Since CaMKII must be in its active form to bind to GluN2B and that this interaction “locks” CaMKII in an active conformation {Bayer, 2001 #955}, we feel that this set of results does address the reviewer’s request. If the reviewer insists that we perform further biochemical experiments, we will be forced to withdraw the manuscript, since my lab is not currently capable of such experiments.

3. The control experiment for no LTP is problematic. Knocking out NMDA receptors increases synaptic strength, as the authors showed previously. Furthermore, NR2B, which the authors propose in their model to be the scaffolding protein for active CaMKII, is not present. Thus, multiple changes occur in this experiment, of which the authors pick one as support for their claim that LTP is missing. The authors need to show how this control prevents the formation of the 'memory-CaMKII'. It seems that this is not possible with the documented experiment as synaptic GluN2B is removed, which is proposed to be the candidate interactor/scaffold of the 'memory-CaMKII'.

The reviewer has a very good point. We had planned to address this issue by comparing the effect of deleting GluN1 to that of replacing GluN1 with a pore dead mutant form of GluN1 from Lonnie Wollmuth. If the pore dead mutant recues the defect our hypothesis would have to be discarded. Alternatively, the failure to rescue would argue for the need for a functional NMDA receptor. This negative result would be couple to immunocytochemistry to ensure that the GluN1 pore dead mutant colocalizes with PSD-95. Unfortunately, we have worked on this experiment for over six months. It has involved generating GluN1 forebrain KO mice and because of the pandemic and numerous technical problems we have been unsuccessful. We suggest two possible alternatives. First, we include the data and explain explicitly that an alternative explanation for the results is the loss of the GluN2B C-tail (current version). Or, if the reviewer insists, we can remove this data from the manuscript.

4. Figure 4 and S4: The run-up of the baseline has a similar slope as the increase after LTP induction, which is problematic. Peptide experiments should be performed with fEPSP recordings with 'solid' baseline rather than 3-point 'baseline'. In essence, it would be helpful to have different LTP protocols to show the same effect. Using cell-permeable peptides does not require whole cell recordings. With fEPSP, the issue of LTP wash-out as a rationale for the short baseline can be circumvented.

This is an important issue. We have studied LTP with whole cell recording for many years and there is definitely a trade off in terms of how much baseline one can get and the rate at which LTP washes out. In most of our experiments we carry out interleaved control experiments to assess the stability of the recordings. (Typically, there is a run up of ~30-50%) In other cases we use a control pathway to monitor stability. In the experiments in Figure 4 we have a control pathway. Both pathways show the same run up prior to LTP, but the control pathway remains stable during the recording until the peptide is applied. This, to me, is a tightly controlled experiment. In Figure S4 the experiments with the 20 μm Myr-AIP were obtained immediately after those with the lower dose, so they are obtained under the same conditions. As discussed above we have repeated some of our key experiments with field potential recording in which longer baselines are recorded.

5. Figure 6: Compared to Figure 4, the LTP in control is rather reduced and not enhanced in paAIP2. It is not uncommon that effect sizes vary, but here it is used as the single line of evidence to indicate that the peptides create more 'room' for later LTP. The experiment needs to be better controlled, e.g. with interleaved experiments.

The reviewer is correct in saying that the magnitude of LTP varies considerably among groups of experiments. Traditionally this variability is controlled for by carrying out interleaved experiments. In the present series of experiments, we have actually used simultaneous recordings from control and transfected cells in the same slice. I have studied LTP for 40 years and it is my belief that this is the most controlled experiment possible. I am not aware of other labs using simultaneous recordings to study LTP.

Revisions Requiring Text Redaction6. The peptides used to perturb CaMKII function are probably among the best tools currently available. However, their specificity remains an issue. They have never been tested systematically but only on a handful of targets. CaMKIINtides have other 'structural' effects than the one on NR2B binding, i.e. interactions with other proteins, and AIP peptides are known to have effects on processes that are not CaMKII-related (Pellicena and Schulman – Front Pharmacol 2014). This issue should be included in the discussion as a general caveat.

Point well taken. We now mention that caution should be used in interpreting our results with these peptides (end of the first paragraph of the discussion).

7. There are no data to show that LTP accumulates during life. Such evidence would be important to support the model propose. Age and LTP 'experience' are not generally applicable criteria when switching back and forth between cultured and acute slices. In essence, one would not expect 'memories' in slice culture, but the effects of the peptides appear to be the same in the two culture systems. With regard to the NMDAR KO experiment, it is then argued that it prevents the formation of in vivo memories and so that no effect of the peptide inhibitors is seen. Thus, whether in vivo or in vitro, synapses appear to 'learn' the same amount. The line of logic in the discussion of these data is not entirely convincing. Given that showing that the corresponding synapses have been LTPed in vivo is highly non-trivial, it is suggested that alternative models and explanations are discussed.

This is an important issue and we agree with the reviewer that further discussion is warranted. I should point out that Lisman {Sanhueza, 2013 #2828;Lisman, 2017 #2794} has asserted that one of the predictions for LTP to be a memory molecular is that there is constitutive CaMKII activity due to prior LTP while the animal was alive. We agree with the reviewer and suggest that “acquired” is a better word than “accumulates”. What we would propose is that NMDAR-dependent Hebbian processes occur early on as neuronal networks are being established. We suggest that this process is occurring by the time that slice cultures are made and are maintained during the time in culture. The issue of whether the basal CaMKII represents prior LTP has been raised throughout. We have done the best that we can to try to link the basal activity to prior LTP. We would welcome any further suggestions to strengthen (or disprove) this link. However, since the link isn’t perfect, we have decided to reorganize the paper so that we first present all of our data characterizing the role of CaMKII in maintaining basal synaptic transmission and then present the data on LTP.

8. Generally, and in the context of the point above, the reviewers suggest to substantially tone down the claim that a mechanism of memory is identified. CaMKII has both basic and potentiation-related synaptic function, which all might well overlap.

We are very careful to refer to our results as “synaptic memory”, with the implication that it could underlie behavior memory. We now make this point more explicit in the discussion. In addition, we have reordered the paper, so that the linkage of the basal action of CaMKII to LTP is played down.

9. LTP is synapse specific. Erasing LTP by blocking the memory molecule would thus be expected to be synapse specific as well. However, both control and LTP pathway are affected by CaMKII perturbation. The authors argue that this is due to LTP accumulation during life. The support for this claim is limited. As the authors show, basal synaptic transmission is reduced by the CaMKII inhibitors. Whether this is LTPed synaptic strength is not clear. An alternative explanation would be that during early LTP expression, before consolidation, LTP can be reversed by inhibiting CaMKII. This should be discussed.

We believe that we have addressed this concern in our previous comments.

10. In CaMKII-α KO mice, LTP is impaired but not absent (Silva et al. – Science 1992). Furthermore, CaMKII autophosphorylation was proposed to facilitate LTP induction, but is not essential (Chang et al. – Neuron 2017). These results are consistent with an important role of CaMKII in LTP, but indicate that the magnitude of LTP blockade depends on the induction protocol and is more in agreement with induction and consolidation of LTP rather than long-term memory. This should be discussed.

I am not entirely clear as to how to address this. There is much variability in the field as to how complete the block of LTP is in the absence of CaMKII or its activity. Certainly no one would disagree that CaMKII is essential for most of NMDAR-dependent LTP. However, it is not entirely clear how this issue relates to constitutively active CaMKII.

11. Halt et al. (EMBO J 2012) proposed that memory consolidation requires the CaMKII/NR2B interaction. The effects of the inhibiting peptides after 10s of minutes, as seen in the present study, are consistent with this notion; whether long-term memory is affected, as proposed in the present study, is not shown. The major novelty of the present study would be that a fraction of CaMKII remains constitutively active to store memory. Reversal of LTP in 30-60 min is not necessarily related to long-term memory, and the notion that reduction of basal transmission reflects the same mechanism is not straight forward. This should be discussed. If more evidence in support of the memory-CaMKII idea can be provided (see above), additional experiments in the context of this item are not expected.

I am hesitant to focus on behavioral memory and terms like “memory consolidation” when discussing synaptic physiology. How the field defines “long-term” is open to discussion. In most studies of LTP, measurements are taken out to 45 min. – 1hour.

12. Figure 2: The effect of AIP expression appears less than 50%. A summary point of two different cell populations is incorrect (see e.g. Levy et al. – Neuron 2015; Kim et al. – Neuron 2015). The data need to be plotted differently and an explanation as to why the effect is smaller needs to be attempted.

Point well taken. We have removed the summary points from the scatter plot and have added a bar graph.

13. Figure S5: The effect of myr-CN27 with MK-801 appears smaller than in other conditions. Does activity in slices before recording have an effect?

We are not certain why the depression is somewhat less. We do not think that it is due to activity in the slice prior to the recording, because incubating slices from the outset in APV does not result in a smaller depression.

14. Line 36: It is unclear what data would support the notion that LTP accumulates during life. The corresponding passage should be referenced or rephrased.

This is a criteria put froward by John Lisman in many of his reviews e.g., {Sanhueza, 2013 #2828}. Throughout we have changed “accumulates” to “acquired”.

15. Lines 76-78: This claim is incorrect. Halt et al. proposed a function for CaMKII in memory consolidation.

I am unable to locate this in the text. What page and paragraph in the reviewer referring to?

[Editors' note: further revisions were suggested prior to acceptance, as described below.]

The manuscript has been improved very substantially, but there are several remaining issues that we regard to be important:Data Interpretation and Discussion1. CaMKII Enzymatic Activity vs. Protein Interactions. The firm conclusion that it is continued CaMKII activity rather than CaMKII binding to relevant postsynaptic binding partners (e.g. the C-terminus of GluN2B) is still not justified by the data.

We accept that the crux of the issue is CaMKII Enzymatic Activity vs. Protein Interactions and we also accept that it is impossible to definitively distinguish between these two alternatives, using CN27 and AIP. Therefore, we remove that part of the discussion that argues for “activity” over “protein interactions”. We also removed the paragraph in the Discussion critiquing the Lee et al. results. Instead, we cite the recent Ozden et al. paper (pg. 12, paragraph 2,line 8 and immediately after this citation we state “Thus, at present it is not possible to distinguish between the disruption of GluN2B binding and the blockade of kinase activity.” We reiterate this on pg 14, paragraph 2, last 3 lines. We have removed “activity” from the title and throughout by replacing “CaMKII activity” with “CaMKII” or the “action of CaMKII”). We also abandon the T site/S site model that has featured prominently in the CaMKII literature, but now appears to have lost its value.

Notably, CaMKII binding to GluN2B was shown to be important for LTP (Barria and Malinow, 2005; Halt et al., 2012). Earlier evidence indicates that the CN27-related CN21 peptide reverses LTP by displacing CaMKII from GluN2B and not by inhibiting CaMKII but (Sanchueza et al., 2011). The authors argue that the AIP peptide only inhibits CaMKII activity without displacing it from GluN2B because earlier work showed that addition of untagged AIP during immunoprecipitation of CaMKII does not disrupt co-precipitation of the NMDAR at 20 μm (Leonard et al., 1999). However, although AIP does not disrupt the preformed CaMKII-NMDAR complex (Leonard et al., 1999), expression of it in HeLa cells prevents binding of CaMKII to GluN2B (see Suppl Figure 2 in Murakoshi et al., 2017). In other words, when expressed in intact cells, AIP does inhibit CaMKII binding to GluN2B. This could also be the case when myrAIP is applied to intact cells,. Perhaps in intact cells CaMKII has a tendency to unbind from GluN2B (potentially due to dephosphorylation of T286) and AIP augments this unbinding or prevents re-binding (potentially by inhibiting re-phosphorylation), while in vitro in immunoprecipitates there is no unbinding and AIP cannot displace CaMKII from GluN2B.

Since we are no longer making any special claims that AIP is acting differently from CN27, we don’t think that the issues raised by the reviewer are any longer relevant.

The issue of AIP acting by inhibiting CaMKII activity vs. CaMKII binding to GluN2B is exacerbated by recently published data indicating that S and T site binding of substrates vs. docking proteins for CaMKII might not be separable (Özden et al., 2020: CaMKII binds both substrates 1 and effectors at the active site. bioRxivhttps://doi.org/10.1101/2020.10.25.354241). This work indicates that T and S sites form a continuum and that most – if not all – binding peptides will affect both, kinase activity and binding to anchoring proteins. Furthermore, the authors argue that their myrCN27 acts via CaMKII inhibition and not CaMKII displacement from GluN2B. Their reasoning is that they had a full effect at 1 μm of myrCN27 whereas Sanchueza et al. (2011) reported that 20 but not 5 μm of the related tat-CN21 displaced CaMKII from GluN2B (and that this correlated with a decrease in transmission and reversal of LTP during maintenance phase), when 5 μm tat-CN21 was sufficient to block CaMKII activity and LTP induction presumably because 5 μm is sufficient to block kinase activity. However, myrCN27 is longer than CN21 and the myr tag could promote binding of this peptide to CaMKII, so it could be much more potent than tat-CN21 in displacing CaMKII from GluN2B and already accomplish displacement at 1 uM.

We agree that with myr-CN27 it is not possible to dissociate protein interactions from enzymatic activity, although it is worth pointing out that this is the argument Sanhueza et al. (2011) used to explain why 5 μm CN21 blocked LTP induction (activity), but did not reverse LTP (protein interaction).

Because of the considerations above, it is mandatory that the authors modify the respective discussion and do not dismiss the notion that 1 μm myrCN27 and AIP in general could have acted by displacing CaMKII.

We have removed this from the discussion.

2. In-Vivo-LTP. The authors argue that GluN1 KO prevents LTP expression in vivo and that this experimental approach could therefore test whether previous LTP in vivo is required for the dependence of synaptic strength on CaMKII. Indeed, the results are consistent with this line of argumentation. However, GluN1 KO has multiple additional effects: It increases synaptic strength and eliminates synaptic NR2B expression, which is thought to be the anchor for active CaMKII. Furthermore, multiple forms of synaptic plasticity (e.g. LTP, LTD, scaling, a.o.) occur during the lifetime of an animal, and notably, NMDARs, CaMKIIα, and CaMKIIαβ were reported to be involved in these forms of plasticity. Thus, the interpretation that in GluN1 KO neurons LTP is absent in vivo is reasonable, but given the other consequences of the KO, it is not conclusive.

While the loss of constitutive CaMKII activity following GluN1 deletion is consistent with prior in vivo LTP, the reviewer pointed out an equally valid conclusion. Since CaMKII binding to the NMDAR is essential for its synaptic action, this disruption could explain our result. To distinguish between these alternatives we have replaced GluN1 with a pore dead mutant. This subunit associates normally with GluN2 subunits and traffics normally to the membrane (Amin, et al., 2017). Thus, the only difference between wt synaptic GluN1/GluN2B synaptic receptors and the mutant is that it is pore dead. Thus, if any non-NMDAR sources of Ca^2+^ can activate CaMKII, it would bind the GluN2B and exert its synaptic action. The fact that the pore dead mutant failed to rescue constitutive CaMKII, suggests that in the freely moving animal non-NMDAR sources of Ca^2+^ are incapable of activating CaMKII, at least in terms of synaptic function. If there were to occur, it would degrade the Hebbian mechanism of plasticity. Aside from the NMDA KO experiment, the reviewer continues to express resistance to the proposal that the constitutive basal CaMKII action represents prior LTP when the animal was alive. I believe there is little doubt that CaMKII contributes to synaptic transmission (Sanhueza, 2007;Sanhueza, 2011;Barcomb, 2016;Gouet, 2012:Goold, 2010;Barcomb,2014;Incontro, 2018). We have done everything we can to test whether this constitutive activity is related to prior LTP. If this is incorrect, what do the reviewers propose is the basis for this activity? We welcome any suggestions the reviewer might have to either strengthen the connection or come up with an experiment that will permanently put this idea to rest. We love doing these kinds of Popper experiments. However, we emphasize that throughout the manuscript we use terms such as “suggests” and “is consistent with this proposal”. We make no claims that it is “conclusive”.

To further link the requirement of constitutively active CaMKII to in vivo LTP, the authors use a type of occlusion experiment. They express the photoactivatable AIP in neurons in utero. Then activate AIP for 40 min in slices after which they start recording. Like inhibition while stimulating, the inhibition without stimulation reduced AMPAR EPSC by 50% compared to neighboring control neurons. LTP in the previously inhibited neurons was about 2-times bigger than that in control neurons. The authors argue that the inhibition of CaMKII liberated "space" for more LTP. Given that LTP is saturable, the origin of this increased "space" is not clear. It could be due to a shift of the baseline further away from the ceiling or, as argued, to the reversal of in vivo LTP that is then newly triggered. Again, a suggestive interpretation, but not conclusive.

We are perplexed by the reviewers’ reticence to accept the conclusion of these experiments. The use of occlusion is not a new idea. It has been used by many investigators, and is generally accepted, in a variety of experiments studying LTP. Just one example is the linkage of the enhancement of CaMKII to LTP (e.g., Lledo et al., 1995), which is part of a figure in one of Lisman’s reviews (Lisman et al., Nat. Rev. Neurosci., 2002). It is also the same logic that Sanhueza et al. (2011) provids to conclude that tat21 reverses saturated LTP (end of the Results section).

In a final experiment, the authors report that 1 µM myr-CN27 reverts LTP during the expression phase. The experiment is performed in a two-pathway experiment, in which both the control non-induced pathway, as well as the induced pathway are reduced. Notably, the strength of the LTPed pathway settles at the same strength as the control pathway, indicating that both basal and the new LTPed synaptic strength is reduced. Myr-AIP is reported to have a similar effect at 20 µM, but not 1 µM. While the 1 µM serves as an important peptide control, the effect of the 20 µM is not as efficient as the 1 µM myr-CN27 as after around 60 min, some potentiation is left and EPSCs are not reduced to 50% of baseline. In a single experiment with field EPSP, myr-CN27 also reduces the fEPSPs of both the LTPed and control pathway, but in the time window observed, the responses do not converge. While these results indicate that CaMKII activity during expression is required, they do not necessarily translate into a mechanism selective for LTP, e.g. that the reduction in control pathway is based on in-vivo-acquired LTP. The different outcomes of the myr-CN27 peptide, the myr-AIP peptide in whole-cell recordings, and the myr-CN27 peptide in fEPSP are fully consistent with different mechanisms, e.g. with CaMKII activity being required for the expression of LTP, while structural assembly being required to stabilize basal synaptic transmission, as reported previously. To distinguish between these possibilities, information about on which CaMKII populations the different peptides operate would be important, e.g. whether myr-CN27 dissociates CaMKII from NR2B, or whether myr-AIP or myr-CN27 change the autoactivation of CaMKII.

We agree with the reviewers’ summary of the data and this is exactly the way we present it in the paper. However, we feel that the reviewer is over interpreting the results. First, we do not agree that this data can be interpreted as distinguishing the mechanism underlying CaMKII’s effects on basal synaptic transmission and LTP. We would argue that any difference in effects on baseline transmission and LTP reflect the effectiveness of the peptide, which varies with the depth of the recorded cell as well as other variables. We also explain that the use of peptides with field potential recording is limited because of the poor access of the peptides to the center of the slice. Thus, the fact that the LTPed pathway does not fall below baseline after one hour of peptide application in Figure S5, although it is still declining, cannot, in our opinion, be used to argue for different mechanisms.

In conclusion, the present study identifies an essential role of CaMKII activity for basal synaptic transmission and LTP expression. It is convincingly shown that this relates to differential effects of cell-penetrating peptides. However, the title and abstract appear too far-fetched as the results do not conclusively show that basal synaptic transmission is the product of acquired in vivo LTP. The reviewers urge the authors to address this issue by more conservative wording of title, abstract, and relevant text passages.

The conclusion that “the present study identifies an essential role of CaMKII activity for basal synaptic transmission and LTP expression” is most surprising. Most of the criticism of our manuscript is that it is impossible to distinguish between “activity” and “protein binding”. We agree and have removed “CaMKII activity” throughout the manuscript.